# Low thermal contact resistance boron nitride nanosheets composites enabled by interfacial arc-like phonon bridge

Ke Zhan[1,11], Yucong Chen [1,11], Zhiyuan Xiong [2,12] ✉, Yulun Zhang[1], Siyuan Ding[1], Fangzheng Zhen[3], Zhenshi Liu[4], Qiang Wei[5], Minsu Liu[1,3,6], Bo Sun [1,7], Hui-Ming Cheng [8,9,10,12] ✉ & Ling Qiu [1,7,12] ✉

Two-dimensional materials with ultrahigh in-plane thermal conductivity are ideal for heat spreader applications but cause significant thermal contact resistance in complex interfaces, limiting their use as thermal interface materials. In this study, we present an interfacial phonon bridge strategy to reduce the thermal contact resistance of boron nitride nanosheets-based composites. By using a low-molecular-weight polymer, we are able to manipulate the alignment of boron nitride nanosheets through sequential stacking and cutting, ultimately achieving flexible thin films with a layer of arc-like structure superimposed on perpendicularly aligned ones. Our results suggest that arc-like structure can act as a phonon bridge to lower the contact resistance by 70% through reducing phonon back-reflection and enhancing phonon coupling efficiency at the boundary. The resulting composites exhibit ultralow thermal contact resistance of 0.059 in$^2$ KW$^{-1}$, demonstrating effective cooling of fast-charging batteries at a thickness 2-5 times thinner than commercial products.

Two-dimensional (2D) materials, such as graphene and hexagonal boron nitride nanosheets (BNNSs), have emerged as promising materials for studying nanoscale thermal transport and developing advanced thermal management solutions[1,2]. The atomically thin nature of 2D materials gives them thermal properties that differ significantly from bulk materials, including dimensionality effects, isotope effects, and phonon boundary/edge scattering[3,4]. Of particular interest for thermal management applications is the anisotropic thermal conductivity of 2D materials. The highly efficient heat conduction in the basal plane compared to across the plane makes them well-suited for heat spreader applications. For example, 2D material-based films have already found commercial applications as heat spreaders in mobile phones. These include highly thermal conductive graphene films (in-plane thermal conductivity of ~1500 W m$^{-1}$ K$^{-1}$) and electrically insulating, low dielectric BNNS films (in-plane thermal conductivity of ~70 W m$^{-1}$ K$^{-1}$) featuring low interference with signal transmission[5–7].

[1]Shenzhen Geim Graphene Center (SGC), Tsinghua-Berkeley Shenzhen Institute (TBSI) & Tsinghua Shenzhen International Graduate School (TSIGS), Tsinghua University, 518055 Shenzhen, China. [2]School of Light Industry and Engineering, South China University of Technology, 510614 Guangzhou, China. [3]Monash Suzhou Research Institute (MSRI), Monash University, 215000 Suzhou, China. [4]Sunwoda Electronic Co., Ltd., 518108 Shenzhen, China. [5]Vivo Mobile Communication Co., Ltd., 523860 Dongguan, China. [6]Foshan (Southern China) Institute for New Materials, 528200 Foshan, China. [7]Institute of Materials Research, Tsinghua International Graduate School, Guangdong Provincial Key Laboratory of Thermal Management Engineering and Materials, Shenzhen, 518055 Guangdong, China. [8]Shenzhen Key Lab of Energy Materials for Carbon Neutrality, Shenzhen Institute of Advanced Technology, Chinese Academy of Sciences, 1068 Xueyuan Road, 518055 Shenzhen, China. [9]Faculty of Materials Science and Energy Engineering, Shenzhen University of Advanced Technology, 291 Louming Road, 518107 Shenzhen, China. [10]Shenyang National Laboratory for Materials Science, Institute of Metal Research, Chinese Academy of Sciences, 72 Wenhua Road, 110016 Shenyang, China. [11]These authors contributed equally: Ke Zhan, Yucong Chen. [12]These authors jointly supervised this work: Ling Qiu, Zhiyuan Xiong, Hui-Ming Cheng. ✉e-mail: xzyscut@scut.edu.cn; cheng@imr.ac.cn; ling.qiu@sz.tsinghua.edu.cn

Despite such a high thermal conductivity, the use of 2D materials as thermal interface materials (TIMs) has been limited by heat conduction across the interfaces between them and substrates (e.g., chips or heatsinks), which may limit the overall thermal transfer[8,9].

TIMs play a vital role in reducing the thermal resistance between two surfaces in many fields, including renewable energy, automotive engineering, and aerospace[10–12]. In fact, the development of high-performance TIMs is much needed to significantly improve existing electronic devices. The overall thermal resistance of TIMs is composed of two parts, the thermal resistance of the materials and thermal contact resistance[13]. 2D materials are highly promising nanoscale thermal fillers in polymer composite films for TIM applications due to their exceptional thermal properties. To maximize the effect of the anisotropic thermal properties of 2D materials, the nanosheets must be perpendicularly aligned in the composite films to give a high thermal conductivity of 10 ~ 70 W m$^{-1}$ K$^{-1}$ [14–16]. However, the thermal contact resistance between TIMs and substrates (e.g., chips or heatsinks) can be 3–5 times the thermal resistance of the material and thus be the main obstacle for practical use[17,18]. Numerous efforts have been made to reduce the thermal contact resistance by incorporating a soft middle layer between the 2D-material TIMs and the substrates[17,19], but the complicated fabrication procedures involved make scalable production difficult, and the improvement is limited. Moreover, the interfacial phonon transport in these studies remains poorly understood due to the existence of multiple interfaces among the 2D materials, polymer matrix, and substrates. Recent studies by Chen, Volz and co-workers have shown that the interfacial phonon transport among 2D nanosheets could be mediated by their stacking structures and binding modes[20–23]. Given the distinct phonon transport behavior that has been extensively reported for 2D materials[4,24], we need to understand how these nanoscale thermal properties can be translated into bulk systems, and in particular, how the microstructure of 2D-material TIMs affects interfacial phonon transport, leading to a low thermal resistance for practical applications.

We report an interfacial phonon bridge strategy to reduce the interfacial thermal resistance of 2D material-based polymer composites for TIM application. We have developed a simple and scalable stacking-cutting method to induce rotation of the BNNSs closest to the cutter in a viscoplastic matrix. This leads to an ultrathin layer of horizontally aligned nanosheets on the cutting surface with obliquely arranged nanosheets beneath the cutting surface. The arc-like arrangement structure of BNNSs significantly reduces the thermal resistance of the resultant composite films by 70% compared to the one without the arc-like structure, giving a low thermal contact resistance of 0.059 in$^2$ W K$^{-1}$, outperforming state-of-the-art dielectric TIMs. Along with the high dielectric strength of to 20.95 kV mm$^{-1}$, the BNNS-based TIMs (BNNS-TIMs) demonstrate the heat dissipation application under a high electric field, such as effectively cooling the fast-charging of batteries. By a non-equilibrium molecular dynamics (NEMD) simulation, we discovered that the oblique BNNSs bridged the phonon spectra mismatch between nanosheets aligned parallel and perpendicular to the film, leading to a significant reduction in the possibility of back-reflection of phonons at the boundary and an improved efficiency of phonon coupling. Our work broadens the understanding of the thermal properties of bulk 2D-material assemblies and provides insights into designing high-performance TIMs to address the challenges of thermal dissipation in modern electronic devices and other industries.

## Results and discussion
### Preparation and structural characterization of BNNS-TIMs
Recently, BNNS has been increasingly used in insulating TIMs because of their high thermal conductivity (up to 750 W m$^{-1}$ K$^{-1}$) and superior dielectric properties (~6.2 eV bandgap)[25]. Unlike graphene where both electrons and phonons serve as thermal carriers, in BNNS, phonons are the sole thermal carrier. BNNS has therefore been used as a model to understand interfacial phonon transport in 2D materials for TIM applications. We used BNNSs with a thickness of ~2 nm and a lateral size of ~1 μm. The BNNSs were mixed with an acrylate polymer to create a compatible dispersion through the strong molecular interactions between the polarized B–N bond and the polymer and then blade-coated to form composite films. BNNS-TIMs were fabricated from BNNS-based composite films by stacking and cutting, by which the advantageous in-plane thermal properties of nanosheets could be utilized for thermal conduction in the through-film direction (Fig. 1A). Hundreds of layers of films with a thickness of ~50 μm were stacked in a mold and then hot-pressed to obtain bulk laminates, where the individual layers were bonded by the melted polymer. The infrared spectroscopy analysis (Fig. S1) suggested the non-covalent interaction between BNNS and the polymer[26,27]. A precision cutting machine was then used to slice the bulk laminate into thin-film BNNS-TIMs with a uniform thickness.

We investigated the structure of the BNNS-TIMs films. Similar to other reported results, perpendicularly aligned BNNSs are found in the middle part of the thin films[28–30] (Fig. 1B), but an ultrathin layer of horizontally aligned BNNSs was observed on the top and bottom cut surfaces of all samples (Fig. 1C–F) owing to the shear force during cutting. Moreover, beneath the cutting surface, BNNSs were observed to rotate at a low loading of 50 wt% (Fig. 1D), but the effect diminished as the filler loading increased to 70 wt% and nearly disappeared at 90 wt%. (Fig. 1E, F).

The BNNS alignment in the composite films was also characterized using X-ray diffraction (XRD). Because the X-rays penetrate a few micrometers[31], exposure of the lateral section of the films to the X-ray yields sharp (100) peaks for all samples, confirming the alignment of BNNSs perpendicular to the film surface. Notably, the (002) peaks exist for all specimens from 50 to 90 wt% BNNS loadings, implying the presence of horizontally aligned BNNSs near the surface region. However, the intensity ratio of (002) to (100) peaks significantly reduces from 3.45 to 0.01 as the BNNS loading increases from 50 to 90 wt%, indicating weakened rotation of BNNSs at high loadings that is consistent with scanning electron microscope (SEM) observations. The average orientational angles derived from the XRD patterns further quantitatively express the trending of BNNNs' rotation (Figs. 1G, S1 and Table S1), where BNNS-TIMs with loadings of 60–90 wt% show larger average orientational angles than the ones at 50 wt% loading (75°–85° vs. ~60°)[32].

### The production of shear-induced nanosheet rotation at the cutting surfaces
The tensile stress-strain behavior of the BNNS/polymer films was measured to understand the mechanism of nanosheet rotation near the film surface. As shown in Fig. 2A, the tensile stress-strain curves of composite films with 50 and 60 wt% BNNS loadings undergo obvious yield and strain softening as the tensile strain exceeds ~10% and reaches the fracture limit at 15–20%. This is a type of ductile fracture behavior that originates from the soft nature of the polymer matrix. When the BNNS loading increases to 70–90 wt%, the fracture limits of the composite film significantly decrease to strains of 3–7%, indicating brittle fracture behavior that has often been observed in polymer/nanoparticle composites at a high nanoparticle loading[33,34]. The ductile to brittle transition is confirmed by the fracture morphology observed from SEM. The fracture surface of composite films with 50 wt% BNNSs are jagged with a width of ~50 μm, and the key feature of ductile fracture is almost 45° shear deformation is presented. In contrast, the fracture of the composite films with 90 wt% BNNSs shows typical brittle fracture features with a straight crack and narrow width of ~10 μm (Fig. 2B). Such a ductile to the brittle transition of the fracture behavior of composite films with an increasing BNNS loading was also supported by their

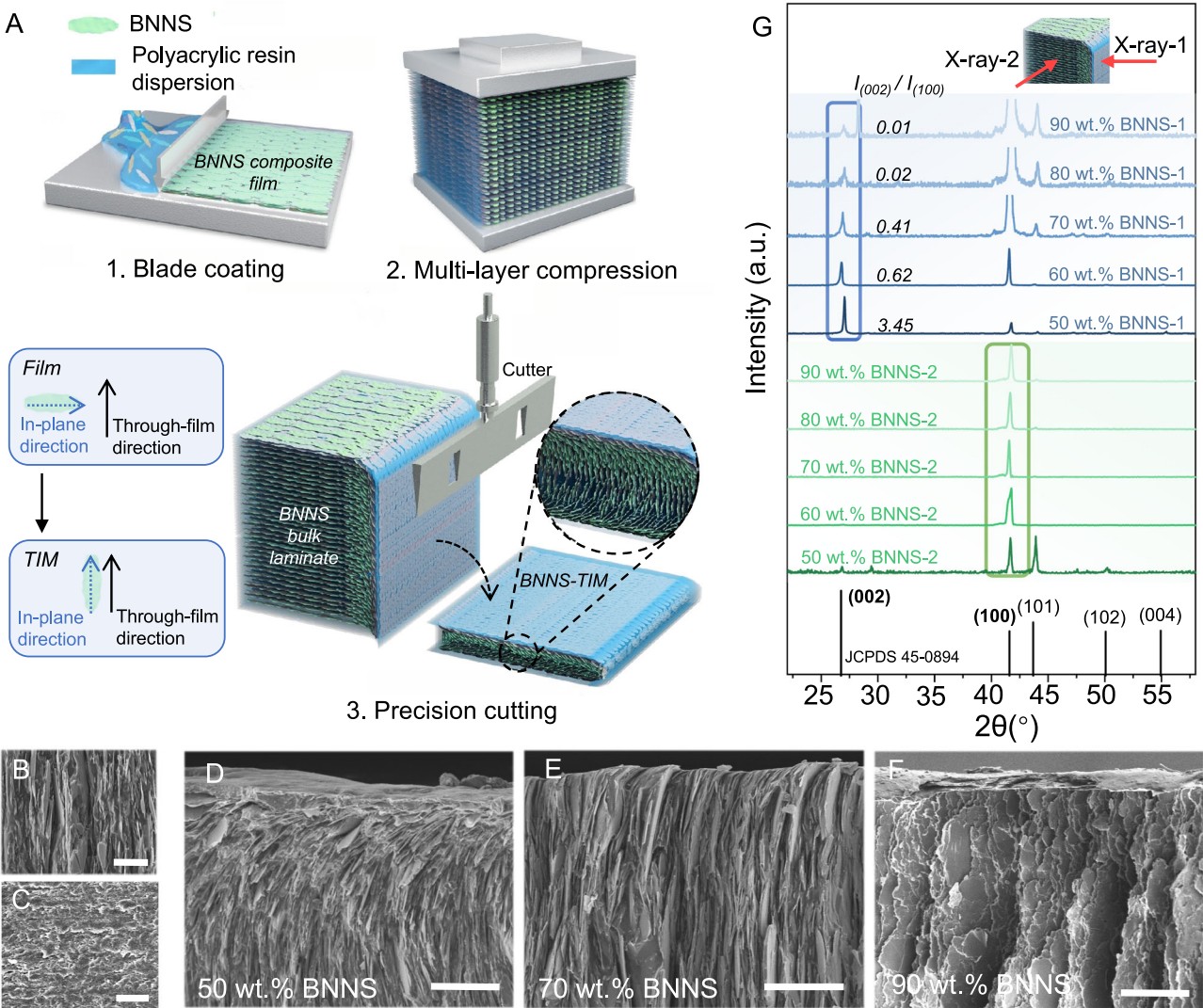

**Fig. 1 | Fabrication and microstructure of BNNS-TIMs. A** Illustration of the stacking and cutting procedures for fabricating BNNS-TIMs, and a schematic depiction of the microstructure of the filler arrangement. **B**–**F** SEM images of BNNS-TIMs: **B** BNNS in the bulk material; **C** view of the machined surface; **D**–**F** side-view SEM images of BNNSs near the cut surfaces of specimens with 50, 70, and 90 wt% loadings. The scale bars in **B** and **C** are 5 µm, and those in **D**–**F** are 20 µm. **G** XRD patterns of specimens with loadings of 50–90 wt.%. X-ray-1 denotes X-rays incident on the cut surfaces of specimens and gives the BNNS-1 patterns, and X-ray-2 refers to those incident on the cross-sections of specimens and gives the BNNS-2 patterns.

viscoplastic rheological behavior and compressive stress–strain curves (Fig. S2).

The observed ductile behavior of the composite films is suggested to play a crucial role in the nanosheet rotation near the film surfaces because it facilitates the motion and re-arrangement of the BNNSs under the shear force caused by cutting. This is supported by a finite element method (FEM) simulation where specimens with different BNNS loadings (50, 70, and 90 wt%) show an arc-like stress–strain pattern near the cutting surfaces when a linear shear force is applied (Fig. 2C, D). This is consistent with the morphology observed by SEM, where the nanosheets gradually rotate from a perpendicular to a parallel direction to form an arc-like pattern. The larger strain observed in specimens with a 50 wt% loading than 70 and 90 wt% loadings are also consistent with the apparent nanosheet rotation seen by SEM.

It is worth noting that the polymer matrix used possesses a low molecular weight ($M = 384.5$), which is significantly lower than the ones ($M_n \sim 10^5$–$10^6$)[16,29,30] of previously reported polymers (e.g., rubber and polydimethylsiloxane) used in 2D materials-based composite films. This may contribute to the ductile (viscoplastic) properties suitable for

film processing and simultaneously ensures good shape fidelity in the resultant films[35] (Fig. S2). Interestingly, when the bulk laminate with 50 wt% BNNSs were sliced at a temperature below the glass transition temperature ($T_g = -20\,°C$) of the polymer matrix (e.g. when immersed in liquid nitrogen at −196 °C) and no nanosheet rotation was observed since the polymer chains were immobile. These results confirm the key role of viscoplastic deformation in producing nanosheet rotation (Fig. S3). In summary, the viscoplastic behaviors of BNNS-TIMs depend on the type and content of the polymer matrix, thereby governing the rotation of BNNS near the cutting surfaces.

## Thermal properties of the BNNS-TIMs

The thermal properties of the BNNS-TIMs were characterized by measuring the through-film thermal conductivity (namely the thermal conductivity of the films in their perpendicular direction), which was estimated from the thermal diffusivity measured by the laser flash method[36] following standard ASTM E1461 (Table S2). It showed a significant increase from ~15 W m⁻¹ K⁻¹ at 50 wt% loading to ~50 W m⁻¹ K⁻¹ at 90 wt.% loading. These values are close to the in-plane $\kappa$ values of BNNS/polymer films at identical BNNS loadings, which indicates the

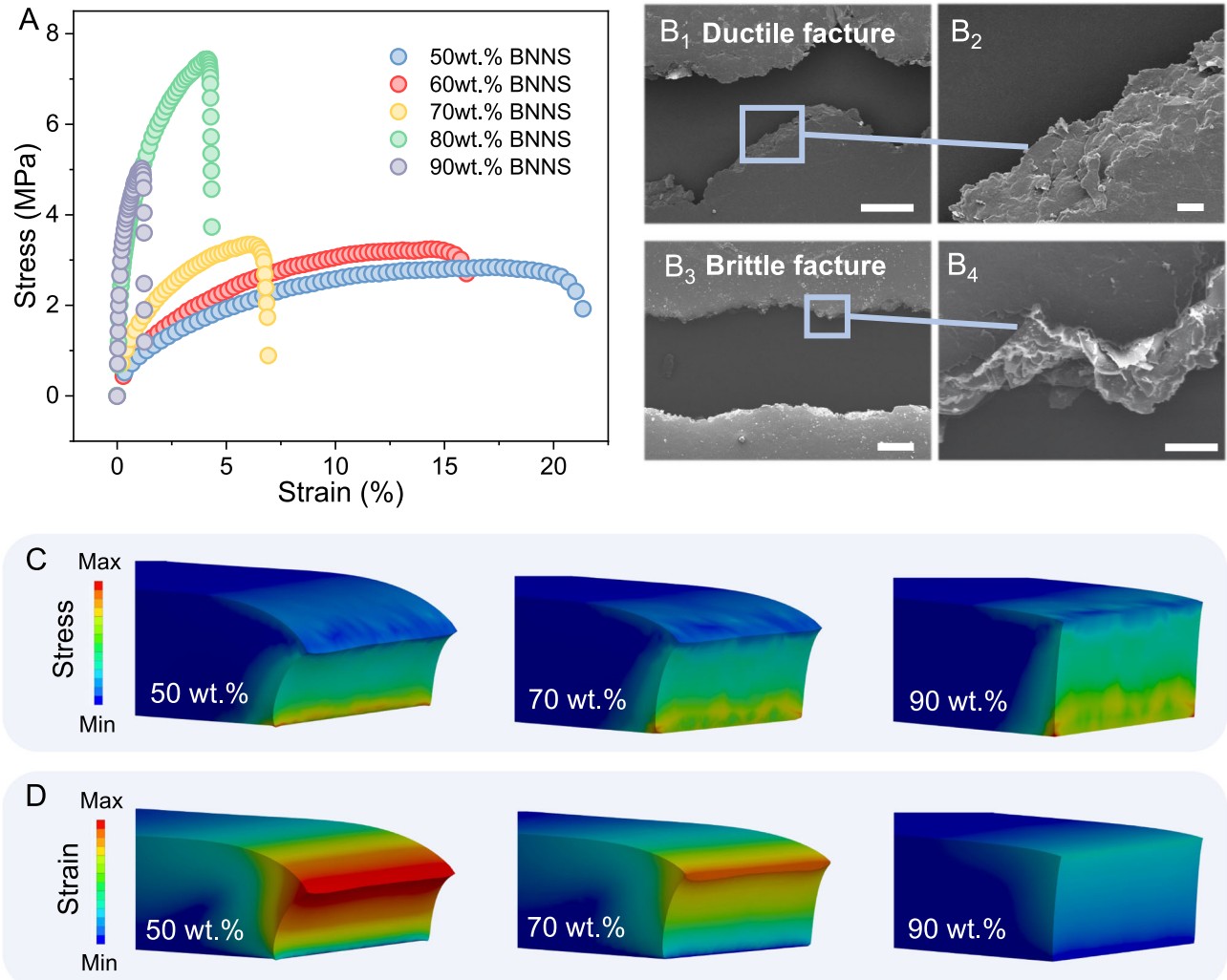

**Fig. 2 | Mechanical properties of BNNS-TIMs. A** Tensile stress–strain curve of specimens with filler loadings of 50–90 wt%. **B** SEM images of the fracture section of specimens with ($B_1$–$B_2$) 50 wt% and ($B_3$–$B_4$) 90 wt.% loadings. The scale bars for $B_1$ and $B_3$ are 200 μm, and for $B_2$ and $B_4$ are 20 μm. **C** and **D** FEM simulation that shows the **C** instant stress and **D** cumulative strain distribution in specimens with 50, 70, and 90 wt% loadings under an external shear force.

efficient reorientation of the filler as a result of the stacking-cutting process (Fig. S4). Moreover, the values are better than other reported studies of hexagonal boron nitride (h-BN) based TIMs by other fabrication methods (e.g., freeze drying, 3D printing), particularly at filler contents over 50 wt% (Fig. 3A and Table S3).

To evaluate the practical performance of the BNNS-TIMs, the total thermal resistance ($R_t$) was measured by guarded hot plate method following standard ASTM D5470, using the equipment with the configuration shown in Fig. 3B. Given the normal packaging pressure for modern electronics (50 psi), the BNNS-TIMs were sandwiched between hot and cold plates made of copper for 10 min to determine $R_t$. Unlike the monotonic change of the thermal conductivity, $R_t$ slightly decreased from 0.55 to 0.49 in$^2$ K W$^{-1}$ as the BNNS loading reduced from 90 to 80 wt%, but there was then an abrupt decrease of around 70% at 70–50 wt% loading, and $R_t$ reached and stabilized at 0.17–0.2 in$^2$ K W$^{-1}$ (Fig. 3B). We normalized the thermal resistance by the film thickness ($d$) to calculate the effective through-film thermal conductivity, $\kappa_{eff} = d/R_t$, which is commonly used to evaluate the thermal dissipation performance of TIMs[37,38]. It was found that the BNNS-TIMs with a 70 wt% content have high $\kappa_{eff}$ values of 9.4 and 12.9 W m$^{-1}$ K$^{-1}$ at 50 and 100 psi pressure, respectively, which are higher than those of reported BNNS-based insulating TIMs and even many carbon-based conductive TIMs (Fig. 3C). BNNS-TIMs with arc-like

structures (50–70 wt% BNNS loading) all give high $\kappa_{eff}$ over 6 W m$^{-1}$ K$^{-1}$, significantly higher than the ones without arc-like structures (80 and 90 wt% BNNS loading) (Fig. S4). It is worth noting that under the normal packing pressure of 50 psi, the $\kappa_{eff}$ (9.4 W m$^{-1}$ K$^{-1}$) of the BNNS-TIMs is considerably higher than some representative commercial dielectric TIMs (<5 W m$^{-1}$ K$^{-1}$) (i.e., Chomerics Gap Filler Pad 976, Honeywell TIP 3500, Henkel TSP 3500).

Apart from the thermal performance, the BNNS-TIMs are also competitive in mechanical and electrical properties and simultaneously possess high dielectric strength of up to 20.95 kV mm$^{-1}$, which can allow their use in heat dissipation for many power electronic devices and power equipment under high electric field, such as fast-charging batteries demonstrated below. As shown in Supplementary Movie 1 and Fig. S5, the composite films (70 wt% BNNS loading) with different thicknesses from 0.2 to 1.0 mm can be repeatedly bent to large angles without cracking or fracturing, showing negligible deterioration in through-film thermal conductivity and demonstrating good mechanical flexibility required for the real-world applications of TIMs. The radar figure (Fig. 3D) shows the overall advantages of our study compared to industrial products and other reports. The softness and thermal conductance are calculated terms from the measured hardness and thermal resistance, for which the details for the comparison are provided in the supporting information (Fig. S6 and Table S4).

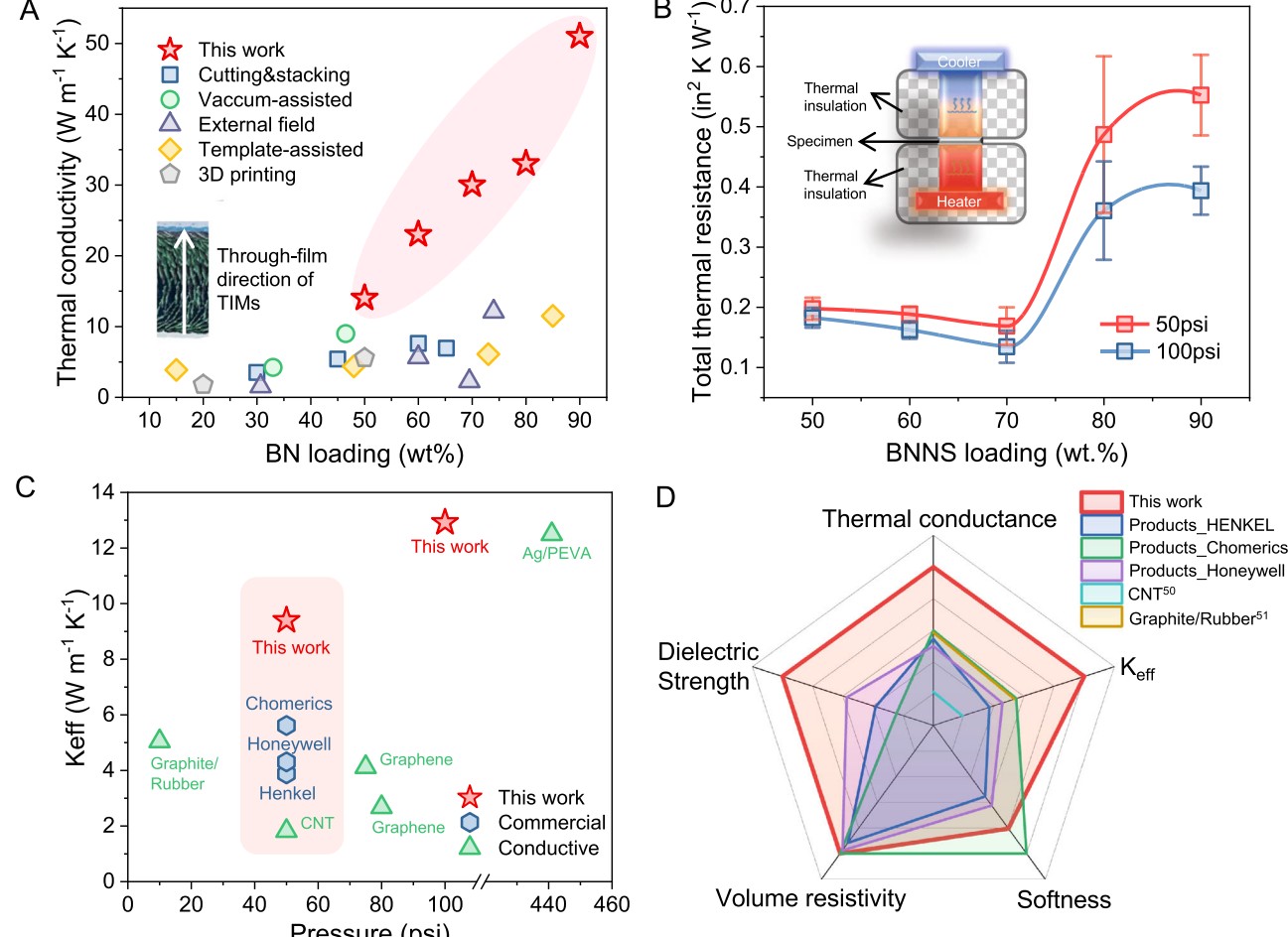

**Fig. 3 | Thermal properties of BNNS-TIMs. A** Summary of the through-film thermal conductivity of h-BN based TIMs using different processing methods[15,16,28–30,61–71]. **B** The total thermal resistance of specimens with 50–90 wt% filler loadings, 1 mm thickness, under a pressure of 50 or 100 psi, with a configuration illustrating the measurement method. At such pressure, the fabricated TIMs can effectively maintain shape fidelity (Fig. S2). **C** Comparison of the effective through-film thermal conductivity under different packaging pressures with commercial products and other reports[72–76]. **D** A radar chart showing the advantages of our BNNS-TIMs compared to other reports and commercial products in various properties[72,73].

## Phonon bridge effect of the arc-like structure for low interfacial thermal resistance

We also studied the influence of the interfacial microstructure of the TIMs on their thermal resistance. According to the equivalent thermal circuit in Fig. 4A, the total thermal resistance ($R_t$) is composed of the material's resistance of the TIMs ($R_m$) and the contact resistance ($R_i = R_{i1} + R_{i2}$) from its two contact surfaces. The material's resistance and the contact resistance can be calculated from the following equations: $R_m = d/\kappa$, $R_i = R_t - R_m$ where $d$ and $\kappa$ are, respectively, the thickness and the through-film thermal conductivity. With increasing BNNS content, its thermal resistance gradually decreases while the contact resistance tends to increase. The ratio of contact resistance to total thermal resistance is 0.36 at 50 wt.% BNNSs and increases to 0.66–0.94 at 60–90 wt% BNNSs, meaning a transition from material-dominated to contact-dominated thermal transfer (Fig. 4A). This gives a minimum value for the total thermal resistance at 70 wt.% BNNSs. We particularly noticed that the thermal contact resistance of the BNNS-TIMs significantly increased 3–4 fold from 0.12 to 0.44–0.52 in² K W⁻¹ when the BNNS content increased from 70 to 80–90 wt%. The great difference in thermal contact resistance between the BNNS-TIMs with 70 and 90 wt% BNNSs is true for thicknesses of 0.2–1.7 mm, as shown in Figs. 4B and S7. Previous studies have suggested that interfacial thermal transport is determined by both surface features of TIMs and phonon coupling[39,40]. Considering that the composite films with 70–90 wt% BNNSs exhibit a consistent surface morphology with a thin layer of horizontally aligned BNNS (Fig. 1C), along with close surface hardness of 92–96 (Shore A) (Fig. 4A) and similar surface roughness of around 5 μm (Fig. S7), it is surmised that the interfacial microstructure of the TIMs plays a key role in the interfacial phonon transport.

The arc-like structure appears in specimens with a filler content of 70 wt% but not in those with a filler content of 80–90 wt%, which provides clues for further explanation assisted by NEMD simulation. Previous studies have reported the influence of the in-plane mis-orientation angle at a grain boundary on the phonon transport of 2D materials from experiments and simulations[41,42]. Similarly, there may be influences from the contact angle between interfacing nanosheets on the thermal transport. Inspired by this, the box in the NEMD simulation was established by placing overlapping BNNSs at a specific contact angle in an attempt to show the influence of filler orientation on the thermal behavior near the interface. And we set the initial strain state of BNNSs in the model as zero since they are negligibly strained from Raman analysis[43,44] (Fig. S1). The hot and cold baths as thermo-static sources with copper plates attached were placed on opposite sides of the simulation box, and between them, monolayer BNNSs were placed in contact at an angle α as the thermal conduction medium (Fig. 4C). Related details are provided in the Methods section. Thermal conductance, as the reciprocal of thermal resistance, is frequently used in theoretical studies. The derived thermal conductance

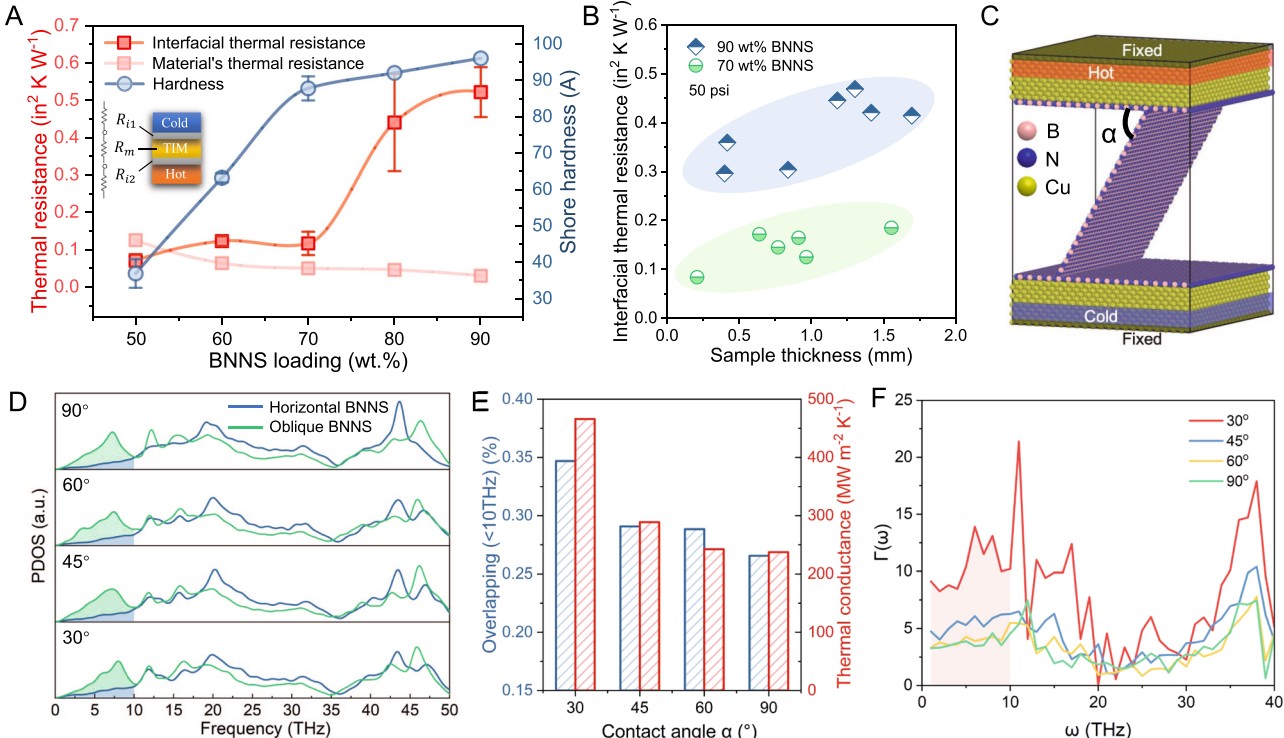

**Fig. 4 | Experimental and simulation proof of the efficacy of interfacial phonon bridge strategy. A** Thermal resistance of specimens with a thickness of 1 mm at filler loadings of 50–90 wt% under a pressure of 50 psi. The red and pink curves, respectively, represent the thermal contact resistance between TIMs and copper plates ($R_i$) and the material's thermal resistance ($R_m$). The blue curve describes the hardness (Shore A) of the TIMs. **B** Thermal contact resistance of specimens with thicknesses in the range 0.2–1.7 mm with loadings of 70 and 90 wt.%. **C–F** NEMD simulation that shows the mechanism of how the filler contact angle $\alpha$ influences the interfacial thermal conductance. The contact angle was set as 30°, 45°, 60° or 90°. **C** Illustration of the simulation box, where the hot bath and cold with copper plates attached are placed on the top and bottom of the box, and BNNSs interconnect with each other at the contact angle of $\alpha$. **D** PDOS spectra at different contact angles and the edges of contact BNNSs are the focus of the calculation. **E** Degree of overlap of the PDOS spectra and the corresponding interfacial thermal conductance at the contact interface between two layers of BNNS. **F** Phonon transmission function spectrum.

at the contact interface showed a significant decrease from 466.1 to 237.0 MW m⁻² K⁻¹ as $\alpha$ increased from 30° to 90°. To understand this, the interfacial phonon transport behavior was studied by computing the phonon density of states (PDOS) by an autocorrelation function method. The interface area of the contacting BNNSs was focused for the calculation, and the results are presented in Fig. 4D. Previous reports suggest that phonons at a low frequency (<10 THz) are easier to transmit through an interface than high-frequency ones[45,46]. Therefore, phonons at the frequency range of 0–10 THz were focused to compute the degree of overlap of the PDOSs between the lying-down and standing-up BNNSs. Similar to thermal conductance, the degree of overlap increased from 26.6% to 34.7% as $\alpha$ decreased from 90° to 30°, indicating that less phonon energy loss through the interface contributes to the increased thermal conductance at smaller contact angles (Fig. 4E). The results of spectral phonon transmission function are also consistent with PDOS calculation[47–49]. As the phonon transmission function (<10 THz) is notably higher when BNNS are positioned at a contact angle of 30° compared to larger angles, diminishing progressively as the contact angle increases (Fig. 4F).

The understanding of angle-dependent thermal conductance in the context of interfacial phonon transport requires careful consideration of multiple factors such as interface bonding, phonon modes, and interface structure[8,24]. Our further calculation shows that the bonding strength shows similar values for different contact angles[50] (Fig. S8), excluding the influence of bonding energy at the interfaces between BNNSs. Meanwhile, previous theoretical works suggest that phonons propagating along the in-plane direction with large mean free paths (MFPs) (over 100 nm) are likely to dominate the thermal conduction in BNNS[51,52]. In such cases, the in-plane propagated

phonons tend to alter direction through boundary scattering at interfaces between overlying BNNSs. By changing the interface structure, as $\alpha$ decreases from 90° to 30°, the increased thermal conductance could possibly be attributed to the decreased back-reflection of in-plane phonons at the interface. It means that the arc-like structure serves as an interfacial phonon bridge[53]. In combining the NEMD results and the SEM observation on the structures of the 70 and 80 wt% specimens, the abnormal increase of thermal contact resistance for 80 wt% specimens can be attributed to the absence of the arc-like structure, which promotes thermal conduction at the TIM-copper interface.

## Industrial application potential of thermal management in fast-charging battery

It is worth noting that the stacking-cutting process is simple and efficient for the scalable production of BNNS-TIMs (Fig. 5A). A large number of BNNS-TIMs could be continuously manufactured with a cutting resolution of 0.2 mm (Supplementary Movie 2). The fabricated TIMs have a uniform thickness and a smooth surface, indicating great potential for continuous production and industrial application (Fig. S9). To validate the performance of BNNS-TIMs for thermal management in modern electronics, we estimated and compared the cooling performance of the BNNS-TIMs using a custom setup, where the TIMs were sandwiched between a ceramic heater and a heatsink under static pressure (43.5 psi) to simulate the heat dissipation process between a chip and a heatsink in electronics. When the power density was set to 10 W cm⁻², the BNNS-TIMs enabled a temperature drop of 45 °C of the heater surface, from 110 to 65 °C at equilibrium. In comparison, a commercial dielectric thermal pad (Henkel TSP3500)

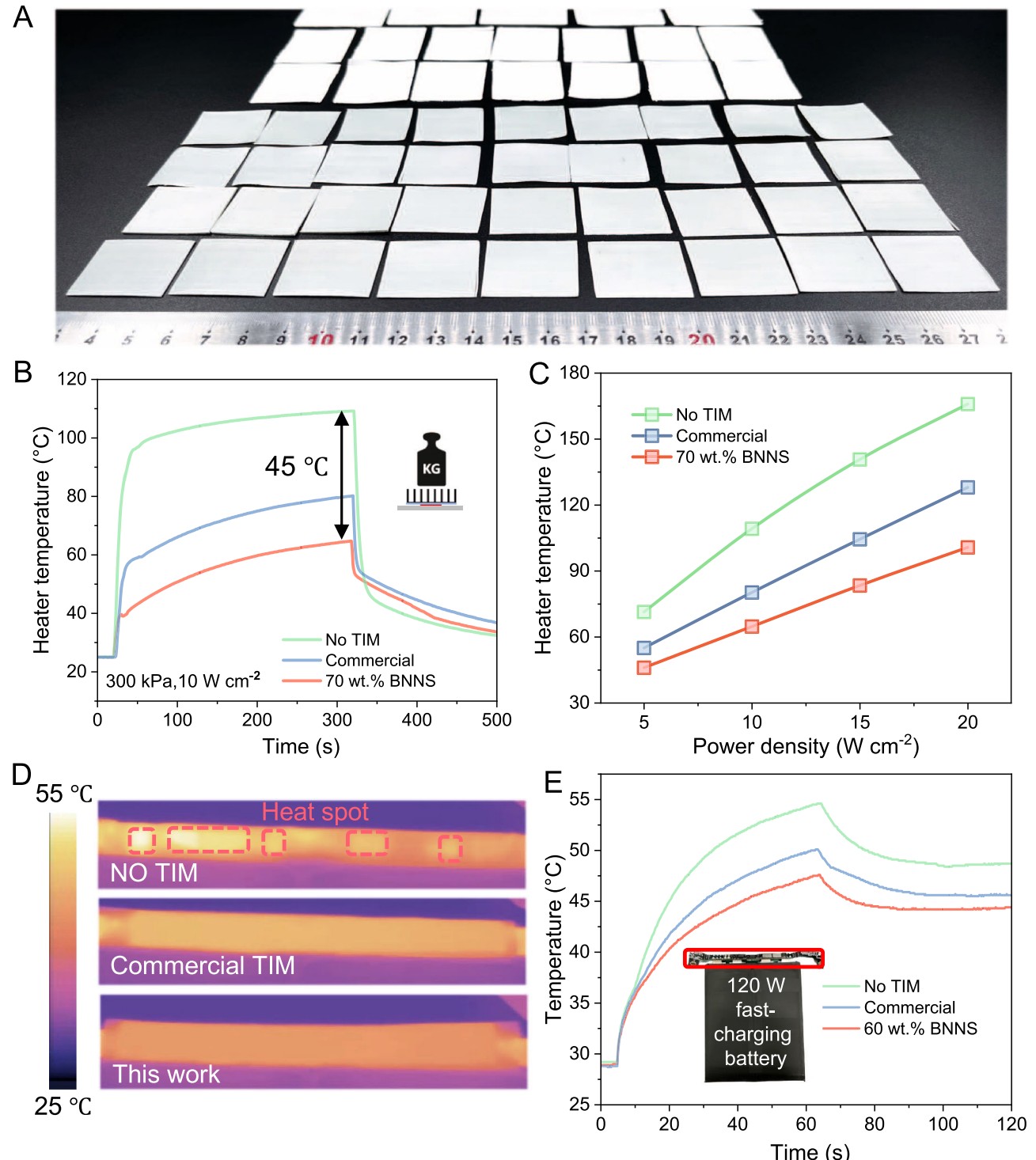

**Fig. 5 | Performance and industrialization of BNNS-TIMs. A** Demonstration of the potential for high scalability in producing BNNS-TIMs using a precision cutting machine. The BNNS-TIMs of 30 × 30 × 1 mm were produced in just 7 s per piece. **B** Temperature–time curves showing the advantage of BNNS-TIMs in heat dissipation under a normal packaging pressure (43.5 psi), with a commercial product (Henkel TSP3500) and no TIMs as references. The heating process lasts for 300 s at a certain power density, followed by ambient cooling. **C** The equilibrium temperature of the heater surface is shown for power density increasing from 5 to 20 W cm⁻² (43.5 psi). **D** and **E** The BNNS-TIMs were used for a fast-charging battery for mobile phones. Infrared images were taken for three different scenarios, without TIMs, with a commercial TIM, and with BNNS-TIMs used after fast-charging for 60 s. The corresponding temperature–time curves for the charging duration were also recorded, and a picture of the 120 W-charging battery is also shown.

showed an inferior cooling performance with an equilibrium temperature of 79 °C at the heater surface (Fig. 5B). The performance was also measured at the power densities in the range of 5–20 W cm⁻², and the BNNS-TIMs produced an additional temperature drop of 9–27 °C compared to commercial products, demonstrating their great potential in high-power electronics (Figs. 5C and S10).

Fast-charging batteries have aroused great interest in portable electronics and new energy vehicles by shortening the charging

period[54,55]. However, the fast charging is accompanied by a much-increased power density, making it challenging to avoid overheated spots and ensure charging safety[54]. At present, commercial TIMs used for cooling fast-charging batteries often have a dielectric strength below 4–10 kV mm⁻¹ and therefore require a minimum thickness of 0.5–1.0 mm for electric insulation. Thanks to the large dielectric strength of BNNS-TIMs over 20 kV mm⁻¹, we successfully demonstrated their use for the thermal management of fast-charging batteries (lithium-ion batteries) at small thicknesses down to 0.2 mm (Table S5). BNNS-TIMs strips with a size of $50 \times 4 \times 0.2$ mm were placed between the chips and thermal spreaders under packaging pressure. The infrared images and the temperature-time curves both confirmed the efficacy of the BNNS-TIMs for thermal dissipation when the battery was charged at a power of 120 W for 60 s followed by 100 W for 60 s. Hotspots were eliminated when our TIMs were used, and the temperature drop was nearly 10 °C from 55 to 46 °C, which was greater than that obtained (5–6 °C) when commercial TIMs (Henkel TSP 3500) were used (Fig. 5D, E). In addition, the thermal stability of the BNNS-based TIMs was proven after eight cycles of temperature shock between 25 and 100 °C (Fig. S11). The small thickness of BNNS-TIMs is beneficial to save space and improve the volume energy density of the battery packs.

In this work, we developed a phonon bridge strategy to fabricate high-performance TIMs based on 2D material/polymer composites. By exploiting the viscoplasticity of the low-molecular-weight polymer matrix, we were able to manipulate the arrangement of 2D nanosheets in the BNNS/polymer composites by sequential stacking-cutting operation, enabling a low thermal contact resistance of 0.059 in² K W⁻¹ with high dielectric strength up to 20.95 kV mm⁻¹. The molecular dynamics simulation shows that the directional phonon scattering in a 2D material leads to angle-dependent interfacial thermal conductance, which can be exploited to improve the efficiency of interfacial phonon coupling and reduce thermal contact resistance. Our simple and scalable manufacturing method enables continuous production of BNNS-TIMs that outperform typical industrial products in a practical scenario, as demonstrated by their use in fast-charging batteries. Overall, this study provides valuable insights into innovative thermal interface design and offers a realistic approach to the industrialization of advanced dielectric TIMs.

## Methods
### Preparation of BNNS-TIMs
BNNSs were produced by our ball milling methods using a glucose solution as medium[5] from raw h-BN powders (PT110, $D_{50} = 45$ μm, Momentive, USA), washed and dispersed in isopropanol. The polymer matrix was an acrylate polymer, and the structural formula is provided in Fig. S1G. The dispersion of BNNS was first blended with the acrylate polymer at low molecular weight ($M = 384.5$) at a rotation speed of 120 rpm for 10 min in a vacuum mixer (MSK-SFM-7, MTI, China). The well-dispersed mixture was then heated at 80 °C to evaporate excess solvent. Next, the mixture with moderate viscosity was gently poured onto the surface of the doctor blading machine (MSK-AFA-H200A, MTI, China) for film molding. Finally, the precursor of the BNNS/polymer film was dried in an oven. A BNNS film of 50 μm thickness was initially cut into pieces of $3 \times 10$ cm. Hundreds of layers were then stacked into a hot-compression mold with a size of $3 \times 10 \times 10$ cm. Followed by compression at 20 MPa and 80 °C for 20 min to strengthen the inter-layer binding using a hot-pressing machine (JHP-600E, Jingsheng Co., Ltd., China). The bulk laminate compressed from hundreds of layers was precisely sliced with a precision cutting machine (SYJ-400, Kejing Auto-instrument Co., Ltd., China) to get BNNS-TIMs with controllable and uniform thicknesses in the range of 0.2–2.0 mm.

## Characterization methods
The morphology and microstructure of the BNNS/polymer films and BNNS-TIMs were examined by scanning electron microscopy (5 kV, Hitachi SU8010, Japan). The degree of filler orientation was characterized by X-ray diffraction with Cu $K_\alpha$ radiation ($\lambda$: 1.54 Å). (XRD, Bruker D8 advance, Germany) in the range 20°–70° at a speed of 5°/min. The details for the calculation of the average angle of the filler orientation are given in the SI. Thermogravimetric analysis (TGA, Netzsch STA 449F3, Germany) was conducted to verify the filler weight fraction in BNNS-TIM. Surface hardness was measured using a Shore hardness tester (LX-A) following the ASTM standard D2240, and hardness mapping was conducted in a Vickers hardness tester (Wilson VH3100, Buehler, USA). Compression tests were conducted using a universal tester (Shanghai XinSanSi CMT-6104, China) equipped with a 10 mm gauge length and operated at a strain rate of 1 mm min⁻¹. The compressive modulus of the samples was taken as the average value of the tangent modulus $E = d\sigma/d\varepsilon$ in the range 0–1% strain, where $\sigma$ is the compressive stress and $\varepsilon$ is the corresponding strain. Tensile tests of BNNS/polymer composite films were conducted in a universal testing system (Instron 5943, USA). The rheologic property of BNNS-TIMs was characterized by a rheometer (Anton Paar MCR 302, USA). Raman spectra were acquired using a high-resolution analytical Raman microscope (Horiba LabRAM HR80, Japan) with 532 nm laser excitation. The bending test was conducted on a flexibility test system (FlexTest, Nanoupe, China). A laser flash appar atus (LFA 467, Netzsch, Germany) was used to study the thermal properties of the thermally conductive pads at 25 °C. The equation for calculating the thermal conductivity was $\kappa = \alpha \times \rho \times C_p$, where $\alpha$, $C_p$, and $\rho$ are the thermal diffusivity, specific heat capacity, and the density of the sample, respectively. The thermal capacity was obtained by differential scanning calorimetry (DSC, TA instrument Q1000, Germany), and the density was measured using the Archimedes method. Thermal resistance was quantified by a thermal interface material tester (LW9389, Longwin, Taiwan), complying with the ASTM standard D5470. The volume resistivity was estimated by an insulation resistance meter (ZC-90, Shanghai Taiou Electronics Co., Ltd., China). Dielectric strength was measured at room temperature by a tester (LK2672X, LANKE Co., Ltd., China) that measured the voltage the material could withstand. It had a testing range of 0–5 kV(AC/DC). To demonstrate the TIM performance for thermal dissipation, infrared images were captured by an infrared camera (TA-60, Shenzhen Dianyang Technology, Co., Ltd., China), and the temperature of the heater was monitored by multi-channel temperature loggers (UT3208, Uni-Trend Technology Co., Ltd., China) with T-type thermocouples.

### NEMD simulation
The size of the simulation box was 80 Å × 80 Å × (80–120) Å, in which the Tersoff potential[56], LJ potential[57], and EAM potential[58] were used to describe the interaction between B–N atoms, B–Cu/N–Cu atoms, and Cu atoms, respectively. Firstly, three dimensions of the simulation box were set to be in the periodic boundary conditions. Then the model was built with BNNSs in linear contact to study the influence of contact mode on thermal conductance by changing the contact angle $\alpha$ (30°, 45°, 60°, and 90°) between two BNNSs. The model was initially relaxed in the NPT process for 100 ps at 300 K to achieve equilibrium. Subsequently, the NVT process was used for further relaxation for 100 ps at 300 K. After that, a few layers of Cu atoms at the top and bottom of the box were fixed for thermal insulation to ensure unidirectional heat transfer. The Cu layer adjacent to the bottom fixed layer was set to be 350 K as the hot source, while several layers of h-BN atoms on the top of the box were set to 250 K as the cold side by accurately rescaling their group velocities. To center the contact between lying-down and standing-up BNNS, the interaction between standing-up BNNS and Cu atoms was neglected. Finally, the system was relaxed in the NVE

process for 1 ns, after which the heat flux, temperature distribution, and velocity per atom were collected. All processes were simulated in large-scale atomic molecular massively parallel simulation (LAMMPS) software[59]. Therefore, the PDOS of atoms at the edge of a BNNS was obtained by the Fourier transform of the velocity autocorrelation function (VACF). The equation is as follows:

$$G(\omega) = \frac{1}{\sqrt{2\pi}} \int_{-\infty}^{\infty} \frac{\langle \mathbf{v}(0) \cdot \mathbf{v}(t) \rangle}{\langle \mathbf{v}(0) \cdot \mathbf{v}(0) \rangle} e^{\mathbf{i}\omega t} dt \qquad (1)$$

where $\omega$ is the phonon frequency, $\mathbf{v}(t)$ is the atom velocity at time $t$. Thus, the degree of overlap at a low frequency (<10 THz) is calculated by

$$S = \frac{\int_0^{\infty} P_A(\omega) P_B(\omega) d\omega}{\int_0^{\infty} P_A(\omega) d\omega \int_0^{\infty} P_B(\omega) d\omega} \qquad (2)$$

where $P_A(\omega)$ and $P_B(\omega)$ represent the PDOS of lying-down and standing-up BNNS atoms at frequency $\omega$.

The interatomic bonding between a pair of atoms at the interface is given by[60]

$$V_{ij} = f_C(r_{ij})[f_R(r_{ij}) + b_{ij} f_A(r_{ij})] \qquad (3)$$

where $i$ and $j$ represent two atoms in two different BNNSs, $r_{ij}$ is the distance between two atoms, $f_C, f_R, f_A$, and $b_{ij}$ are defined in the Tersoff potential. Thus, the interatomic bonding can be calculated through atom coordinates derived from MD data. The average interatomic bonding at the interface of BNNSs contact can be calculated by

$$V = \left\langle \sum V_{ij} \right\rangle \qquad (4)$$

where <> means an average overall pairwise bonding within the cutoff region. The results indicate the interatomic bonding at the interface of BNNSs at contact angles of 30°, 45°, 60°, and 90° are approximately the same (−3.74 to −3.64 eV), compared with in-plane B–N bonding −4.47 eV, which excludes the interference from interatomic bonding on thermal conductance.

The phonon transmission spectrum through the edge of two BNNSs can be described as

$$\Gamma(\omega) = \frac{q(\omega)}{k_B \triangle T} \qquad (5)$$

where $q(\omega)$ is the heat current spectrum, $\Delta T$ is the temperature difference between two different BNNSs, and $k_B$ is the Boltzmann constant. $q(\omega)$ can be calculated by

$$q(\omega) = 2\mathrm{Re} \sum_i \sum_j \int_{-\infty}^{+\infty} dt e^{\mathbf{i}\omega\tau} \left\langle \mathbf{F_{ij}(t)} \cdot \mathbf{v_j}(0) \right\rangle \qquad (6)$$

where $i$ and $j$ represent two atoms in two different BNNSs, $v_j$ is the velocity of atom $j$, $F_{ij}$ represents the interatomic force between two atoms.

### Finite element method simulation
Finite element analysis was used to simulate the cutting process. A finite element model was constructed using the commercial software ANSYS 2019R2 Workbench. The material model was defined based on orthotropic elasticity since the nanosheets in the bulk laminates are highly aligned. The bulk laminate was assumed to be a unified entity with measured mechanical properties incorporated. During the cutting process, the edge of the film was subjected to vertical shear stress

based on the experimental conditions. As the simulation was conducted for static structures, the shear stress was simplified to maintain a constant magnitude and direction. Consequently, the stress and strain conditions near the cutting face after the application of the shear stress were the focus of the research. In the simulation, a film of size 200 μm (L) × 100 μm (W) × 50 μm (H) was created, and a constant shear stress of 0.2 N was applied in the high extension direction, represented by the Z-axis direction. The experimental conditions were kept consistent in order to study the stress and elastic strain conditions of the films with varying BNNS loadings. Considering the potential structural heterogeneity and the change in the mechanical property of the composite films, we also perform FEM models based on isotropic elasticity, the results are shown in Fig. S12.

### Calculation of average orientational angle of BNNSs in composites
The method reported by Uchikoshi et al.[32] was used to quantitatively illustrate the degree of filler alignment in composites with an average orientation angle $\bar{\alpha}$. The angle $\alpha_{hkl}$ is defined as the angle between the crystal plane ($hkl$) and (001), therefore, 0° refers to parallelly aligned BNNSs while 90° denotes perpendicular ones. The $\alpha_{hkl}$ for main crystal planes in the h-BN lattice can be calculated from the following equation:

$$\cos \alpha_{hkl} = \frac{\frac{\sqrt{3}a}{2c} l}{\sqrt{h^2 + k^2 + hk + \frac{3a^2}{4c^2} l^2}} \qquad (7)$$

For h-BN, $a = 0.2173$ and $c = 0.6657$, and the results are shown in Table S1. The average orientation angles $\bar{\alpha}$ for specimens with different loadings were calculated as the sum of multiply of $\alpha_{hkl}$ and the weighted intensity of the corresponding peak of the XRD pattern, as shown in the equation below:

$$\bar{\alpha} = \sum \left( \alpha_{hkl} \times \frac{I_{hkl}}{\sum I_{hkl}} \right) \qquad (8)$$

3–5 specimens were tested for each loading, and the data are presented in Table S1.

### Reporting summary
Further information on research design is available in the Nature Portfolio Reporting Summary linked to this article.

## Data availability
All data needed to evaluate the main conclusions of this research are available within the main text, the Supplementary Information, and the Source Data files. Any additional request on raw data produced in the current study will be fulfilled by the corresponding authors. Source data are included with this paper. Source data are provided with this paper.

## Code availability
The simulation codes used in this study are available from the corresponding authors on reasonable request.

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

## Acknowledgements

Special thanks to Dr. Weiren Fan, Dr. Jiuyi Zhu, Dr. Lanshu Xu, Qinshu Li, Runhua Gao, Wenzhen Xie, and Hualiang Cai. This work was supported by the National Key Research and Development Project (Nos. 2019YFA0705403, 2022YFA1205300, L.Q.), the National Natural Science Foundation of China (No. T2293693, L.Q.), the Guangdong Innovative and Entrepreneurial Research Team Program (No. 2017ZT07C341, L.Q., H.M.C.), the Guangdong Basic and Applied Basic Research Foundation (No. 2020B0301030002, L.Q., H.M.C.), and the Shenzhen Basic Research Project (Nos. WDZC20200824091903001, JSGG20220831105402004, L.Q.). Z.X. thanks the financial support from the South China University of Technology.

## Author contributions

K.Z., Y.C., M.L., Z.X., H.-M.C., and L.Q. did the conceptualization. K.Z., S.D., and M.L. developed the methodology. K.Z., Y.Z., and Y.C. did the fabrication and characterization. Y.C. designed and performed the NEMD simulation, and F.Z. performed the FEM simulation. Z.L. and Q.W. tested the cooling effect of the fast-charging batteries; K.Z., Y.C., Z.X., B.S., and L.Q. analyzed the data, and discussed the mechanism; K.Z., Y.C., Z.X., L.Q., and H.-M.C. co-wrote the manuscript with input from all authors. All authors have given approval to the published version of the manuscript.

## Competing interests

The authors declare no competing interests.
