## [Peer Review File · Nature Communications]

Low thermal contact resistance boron nitride nanosheets composites enabled by interfacial arc-like phonon bridgeREVIEWER COMMENTS

Reviewer #1 (Remarks to the Author):

The authors report in this manuscript the interfacial phonon bridge strategy to reduce the thermal contact resistance of BNNSs based polymer composites. The resultant composite film which is used as TIMs exhibits ultralow interfacial thermal resistance and high dielectric strength. The following questions should be addressed to improve the quality of this manuscript.

1. Page 4, the authors mixed BNNSs with a viscoplastic low-molecular-weight polymer and then blade coated to form composite films, but the composition of this polymer is not mentioned in this manuscript. The authors should provide the type of the polymer clearly.
2. The authors should add more discussion on how to adjust the viscoplasticity of the polymer and how the viscoplasticity of the polymer affected the orientation of BNNS during cutting.
3. Figure 2C and 2D, in the FEM simulation, the authors have defined the material model based on orthotropic elasticity. However, there is a ductile to brittle transition as filler loadings shifted from 50% to 90%. Is it reasonable to define FEM models based on isotropic elasticity?
4. In Table S5, the authors have mentioned that thermal conductivity of BNNS-TIMs at a filler loading of 60 wt.% with different sample thickness is different. Therefore, whether the samples with different filler loadings in Figure 3A are of the same thickness.
5. Figure 3B, given that the material's resistance is calculated with the thermal conductivity, and the contact resistance is calculated by the total resistance in Figure 3B and the material's resistance in Figure 4A. Whether the thickness of the specimens in Figure 3B is all 1mm which is the same as Figure 4A.
6. Figure 4E, the authors have mentioned that by changing the interface structure, as α decreases from 90° to 30° , the back-reflection of in-plane phonons decreases and the thermal conductance increases. Will the thermal conductance further increase when α is smaller than 30° ?
7. Figure 5A, the authors use BNNS-TIMs with a filler loading of 60 wt.% to illustrate the concept of continuous manufacturing. Why not use the optimal sample with a filler loading of 70 wt.%?

Reviewer #2 (Remarks to the Author):

The manuscript reports on a strategy to fabricate thermal interface materials for dissipating the heat in electronic systems. The strategy is to create inclined alignment of BNNS in a polymeric matrix by using the shear force that develop when cutting a BNNS-polymer composite. The manuscript is well written, and the results are interesting. However, alignment or misaligning nanosheets using shear is not a new concept per se and has been reported in other works using other types of nanosheets or 2D particles. Main example of aligning through shear is by tape casting. Furthermore, the authors propose the tilted alignment in an arc as a key mechanism to provide more efficient thermal conductivity, which is true, but aligning BN or BNNS vertically to increase the thermal conductivity has also already been reported in the literature. Finally, the discussion about thermal interface is quite confusing: at times it seems that the authors refer to their entire composite as the thermal interface (being a TIM); at other times it seems that the authors might refer to the actual interface between the composite and the electronic, which would be in my opinion the

interface that should be considered here. In the case it is the last one of these interfaces, then the authors should better characterize the morphology, roughness, thickness, or this interface. Finally, I am confused by the lack of information about the BNNS: their dimensions, how smooth or wrinkled they are, as well as about the polymer: it is apparently a thermoplastic but what is its chemistry? is it binding to the BNNS?

Reviewer #3 (Remarks to the Author):

Review attached.

Article reference: NCOMMS-23-50747

Title: Changing the interfacial structure in boron nitride nanosheets/polymer composites for a low thermal contact resistance using a phonon bridging strategy

Recommendation: Major revision

In this work, the authors demonstrated that the interfacial phonon bridge is an effective strategy to optimize the thermal resistance within the hexagonal boron nitride nanosheets based composite films. With this method, the authors reported that the contact resistance can be significantly reduced by 70%. The results of this work show promising applications in the field of thermal interface materials (TIMs). However, due to the following issues listed below, I suggest the authors should adequately address the following comments to significantly improve this manuscript before potential publication:

1. The recent advances on the thermal properties of two-dimensional materials [*Applied Physics Reviews*, **10**, 041404 (2023); *Physical Review B*, **107**, 165424 (2023)] and interfacial thermal transport [*Nanotechnology*, **33**, 035707 (2022)] are highly related to this work, and thus should be cited in the introduction part to provide a timely survey of literature studies for the readers.
2. The interfacial thermal resistance between Van der Waals stacking 2D materials in the out-of-plane direction is a prevalent issue in thermal management. In fact, the concept of phonon bridging is not new for interfacial optimization, see for instance *Nat. Commun.* **7**, 11281 (2016). I suggest the authors should moderate its tone. The advantage for optimization simply by cutting operation would be a more intriguing focus for this manuscript.
3. The arc-like structure resulting from cutting is observed only at the surface of BNNS-TIMs. How would the optimization performance of thermal contact resistance R vary with the size of the sample? With larger BNNS-TIM structures, the impact of surface

in principle should eventually become negligible. Would this effect potentially diminish the optimization efficiency of R from surface cutting?

4. The authors attempted to use the contact angle to elucidate the morphology dependent on filler content and the resulting changes in R . The second paragraph on page 10 implied that samples with high filler content, such as 90 wt%, exhibit high contact angles associated with lower thermal conductance and higher R . However, Figures 1(f) and 2(c) indicated no angle difference for BNNS at the surface. The provided theoretical explanation may lack accuracy.

5. The authors employed the overlap of phonon density of states (DOS) to explain the interfacial thermal transport. This approach might be oversimplified for comprehending the complicated interfacial thermal transport mechanism, at least for the papers in the *Nat. Commun.* level. The recent advancements in theoretical and computational methods offer numerous alternative approaches to elucidate interfacial thermal transport and provide more detailed insights into phonon transport mechanisms, such as the interfacial phonon transmission spectrum analysis [see for instance, *Nano Letters*, **21**, 2634–2641 (2021)]. I suggest the authors should strengthen the theoretical analysis part.

6. Figure 1(f) also indicated that the BNNS likely exhibit a curved surface after cutting, which may potentially cause a partial shift from out-of-plane thermal transport to in-plane transport. However, the model used in the theoretical explanation did not account for this notable feature.

Response to Reviewer's Comments

Dear Editors and Reviewers,

Thank you for your valuable comments on our manuscript. We have review comments carefully and revised our manuscript accordingly. All comments have been replied as below and revisions are shown in our revised manuscript and corresponding modifications are highlighted in blue.

Reviewer #1

The authors report in this manuscript the interfacial phonon bridge strategy to reduce the thermal contact resistance of BNNSs based polymer composites. The resultant composite film which is used as TIMs exhibits ultralow interfacial thermal resistance and high dielectric strength. The following questions should be addressed to improve the quality of this manuscript.

Response: We are grateful for the reviewer's insightful feedback on our manuscript.

Comment 1. Page 4, the authors mixed BNNSs with a viscoplastic low-molecular-weight polymer and then blade coated to form composite films, but the composition of this polymer is not mentioned in this manuscript. The authors should provide the type of the polymer clearly.

Response: We appreciate the reviewers' comment. The polymer is an acrylates polymer with a low molecular weight ($M_n \sim 10^3 - 10^4$). This information has been updated in the section of Materials and Methods in the main text (Page 14).

Comment 2. The authors should add more discussion on how to adjust the viscoplasticity of the polymer and how the viscoplasticity of the polymer affected the orientation of BNNS during cutting.

Response: Thank you for the reviewer's constructive comment. It is important to clarify that our emphasis was on modifying the viscoplasticity of the composite rather than the polymer itself. To this end, we have experimented with multiple types of polymers in the early stages of this project. After extensive evaluation, we selected an acrylates polymer with a low molecular weight as the optimal polymer matrix. This choice was based on its low glass transition temperature ($T_g = -20^\circ\text{C}$), which allows the composites desirable viscoplasticity and processability at room temperature.

As for tuning the viscoplasticity of the composites, we achieve it by tuning the mass ratio of BNNS to the polymer matrix. It is observed that the viscoplasticity of BNNS-TIMs decreases with reduced content of the polymer (Figure 2A-B). Furthermore, our modeling results show that the viscoplasticity of the composite films can facilitate the motion and re-arrangement of the BNNSs under the shear force caused by cutting (Figure 2C-D). With higher viscoplasticity, the nanosheet rotation appears to readily occur, resulting in obliquely aligned BNNS near the film surfaces (Figure 1D-F). Interestingly, when the composite films with 50 wt.% BNNSs was sliced at a temperature below the glass transition temperature ($T_g = -20^\circ\text{C}$) of the polymer matrix (*e.g.*, when immersed in liquid nitrogen at -196°C), no nanosheet rotation was observed (Figure S3) since the polymer chains were immobile, further confirming the key role of viscoplasticity in producing oblique alignment of nanosheets.

To clarify our strategy on modulating the viscoplasticity of the composite and tuning nanosheet rotation, we have added the sentences “In summary, the viscoplastic behaviors of BNNS-TIMs depend on the type and content of polymer matrix, thereby governing the rotation of BNNS near the cutting surfaces.” in Page 6 of the main text.

Comment 3. Figure 2C and 2D, in the FEM simulation, the authors have defined the material model based on orthotropic elasticity. However, there is a ductile to brittle transition as filler loadings shifted from 50% to 90%. Is it reasonable to define FEM models based on isotropic elasticity?

Response: The reviewer raised a great question. Structural analysis of the composite films reveals that a majority of the BNNS were aligned perpendicular to the middle section, while fewer were obliquely aligned near the film surface (Figure 1B-F). Notably, the perpendicular alignment becomes more pronounced with increasing BNNS loading from 50% to 90% (Figure D-F). Given that the average orientational degree consistently exceeds 60 degrees (Figure S1D), the mechanical properties of the composite film could exhibit directional variability. Consequently, we opted for the use of orthotropic elasticity in FEM simulation. However, considering the potential structural heterogeneity and the change of the mechanical property of the composite films, we also follow the reviewer’s advice and perform FEM models based on isotropic elasticity. The results (Figure R1) shows a similar trend to Figure 2C-D, in cumulative strain pattern as filler loading increase, though with minor variations in strain. Both sets of results support our conclusion in explaining nanosheet rotation behavior under shear cutting.

In the revised version, we have provided the rationale for opting for orthotropic elasticity in the

main text (Page 17) as follows: “The material model was defined based on orthotropic elasticity since the nanosheets in the bulk laminates are highly aligned.” Simultaneously, modeling results based on isotropic elasticity have been incorporated into the Supporting Information as Figure S12 (Page 10).

Figure R1: FEM simulation results based on isotropic elasticity that show (A) instant stress and (B) cumulative strain distribution in specimens with 50, 70, and 90 wt.% loadings under an external shear force.

Comment 4. In Table S5, the authors have mentioned that thermal conductivity of BNNS-TIMs at a filler loading of 60 wt.% with different sample thickness is different. Therefore, whether the samples with different filler loadings in Figure 3A are of the same thickness.

Response: We appreciate the reviewer’s careful observation regarding the consistency of sample thickness. The observed differences in Table S5 could be attributed to many factors such as sample uniformity, instrument sensitivity, and statistical error. Specifically, the thermal conductivity measurement equipment used (Netzsch LFA 467) provides a standard deviation of $\pm 3\%$, stemming from factors like sample geometry¹. Considering the minor variance in thermal conductivity within the range of $17.9\text{--}21.2\text{ W m}^{-1}\text{ K}^{-1}$, these data still affirm the stability of BNNS-TIMs’ performance under continuous manufacturing processes.

We confirm that for the thermal characterization of our BNNS-TIMs presented in the main text, including the results shown in Figure 3A, all samples with different filler loadings were maintained at a uniform thickness of 1 mm to ensure the validity and reliability of our findings. To clarify this, we have added thickness information of BNNS-TIMs in figure caption in the main text (Page 9).

Comment 5. Figure 3B, given that the material' s resistance is calculated with the thermal conductivity, and the contact resistance is calculated by the total resistance in Figure 3B and the material' s resistance in Figure 4A. Whether the thickness of the specimens in Figure 3B is all 1 mm which is the same as Figure 4A.

Response: As we respond in comment 4, all specimens for thermal measurement were indeed 1 mm in thickness, unless we mentioned different thicknesses in the samples.

Comment 6. Figure 4E, the authors have mentioned that by changing the interface structure, as a decreases from 90° to 30°, the back-reflection of in-plane phonons decreases and the thermal conductance increases. Will the thermal conductance further increase when angle is smaller than 30°?

Response: Thanks for the reviewer's insightful comment. In response to the reviewer's question, we have extended our investigation through additional MD simulations for BNNSs at a contact angle of 15°. The results are detailed in Figure R2A. Although the setting angle between the main parts of the BNNSs is 15° in our model, the actual contact angle is slightly larger due to the repulsive forces between the BNNSs, causing the contact part of the tilted BNNS's atoms to orient more perpendicularly to the horizontal BNNS. Our simulation shows that the thermal conductance at this 15° angle is 543.6 MW m⁻² K⁻¹ for the hot side and 490.8 MW m⁻² K⁻¹ for the cold side, as shown in Figure R2B. This represents a modest increase of about 10% compared to the 30° angle. The new results have been updated in the Supporting Information as Figure S8B&D (Page 8).

We didn't further reduce the contact angle in the modelling. As the contact angle approaches 0°, a substantially larger quantity of nanosheets would be necessary to construct the arc-like phonon bridge facilitating thermal transport across interfaces between the device and TIMs. This situation differs significantly from our experimental conditions, where contact angles above 30° are commonplace. Additionally, attempting simulations at angles less than 15° proved challenging, as mentioned earlier; the system is difficult to stabilize. The broad angle change from 90° to 15° can effectively supports the observed angle-dependent thermal transport of 2D BNNS.

Figure R2: (A) Illustrates the MD model, highlighting that the actual contact angle between parallel and tilted BNNS is greater than the preset angle of 15°. (B) is an addition for Figure S8B, displaying the thermal conductance between BNNS at varying contact angles of 15°, 30°, 45°, 60°, and 90°. The conductance values are shown adjacent to the hot side (red bars) and the cold side (blue bars).

Comment 7. Figure 5A, the authors use BNNS-TIMs with a filler loading of 60 wt.% to illustrate the concept of continuous manufacturing. Why not use the optimal sample with a filler loading of 70 wt.%?

Response: Our initial choice of the 60 wt.% filler loading was primarily driven by its manufacturability, making it an ideal candidate to demonstrate the concept of continuous manufacturing. However, recognizing the reviewer's point about optimal thermal performance, we have now incorporated additional data for the 70 wt.% filler loading sample. As presented in Movie S2, this higher filler loading sample was successfully produced in a large-scale manufacturing setting. The samples were fabricated in dimensions of 10 × 5 mm and offered in thickness variations of 0.2, 0.5, and 1.0 mm. Importantly, both the thermal and dielectric performance of the 70 wt.% sample align with the data in the main text, as detailed in Table R1. This further highlights the versatility and scalability of our manufacturing process. Table S5 has been replaced by the content from Table R1, and updated in Supporting Information (Page 13).

Table R1. Physical parameters and electric/thermal performance of BNNS-TIMs with a filler loading of 70 wt.% obtained by continuous manufacturing.

Sample Thickness (mm)	Density (g cm ⁻³)	Breakdown voltage (kV)	Dielectric strength (kV mm ⁻¹)	Total Thermal resistance (K in ² W ⁻¹)
0.2	1.605	6.2	31	0.042
0.5	1.631	7.9	15.8	0.070
1.0	1.636	7.8	7.8	0.121

Reviewer #2

The manuscript reports on a strategy to fabricate thermal interface materials for dissipating the heat in electronic systems. The strategy is to create inclined alignment of BNNS in a polymeric matrix by using the shear force that develop when cutting a BNNS-polymer composite. The manuscript is well written, and the results are interesting. However, alignment or misaligning nanosheets using shear is not a new concept per se and has been reported in other works using other types of nanosheets or 2D particles. Main example of aligning through shear is by tape casting. Furthermore, the authors propose the tilted alignment in an arc as a key mechanism to provide more efficient thermal conductivity, which is true, but aligning BN or BNNS vertically to increase the thermal conductivity has also already been reported in the literature.

Response: We are grateful for the reviewer's acknowledgement of our writing and interesting findings. We agree with the reviewer's comments regarding prior strategies utilizing shear to align nanosheets and the significance of vertical alignment of BNNS in enhancing thermal conductivity, as discussed in our introduction. However, the overall thermal resistance of thermal interface materials (TIMs) encompasses not only the inherent thermal resistance of the materials but also the thermal contact resistance. Although vertical alignment of BNNS can enhance the thermal resistance of the materials themselves, it may introduce substantial thermal contact resistance between TIMs and the substrate due to directional phonon transport along 2D nanosheets. In this work, we focus on optimizing interfacial thermal transport by tilted BNNSs contact, distinct from strategies employing solely the vertically aligned nanosheets. While we employ previously reported fabrication approach mentioned by the reviewer (*i.e.*, layer stacking and shear cutting) to tune the orientation of BNNS, our key discovery is the contact-dependent thermal resistance and firstly demonstrate how this nanoscale phonon bridge effect can be translated into bulk systems to

enable high-performance TIMs. Overall, this study provides expands the knowledge of the thermal properties of bulk 2D-material assemblies and offers a realistic approach to the industrialization of 2D materials-based dielectric TIMs.

Finally, the discussion about thermal interface is quite confusing: at times it seems that the authors refer to their entire composite as the thermal interface (being a TIM); at other times it seems that the authors might refer to the actual interface between the composite and the electronic, which would be in my opinion the interface that should be considered here. In the case it is the last one of these interfaces, then the authors should better characterize the morphology, roughness, thickness, or this interface.

Response: We apologize for any confusion regarding the term 'interface' as used in our manuscript. In general industry practice of TIMs, 'interface' refers to any component situated between the heater and the heatsink in an electronic system. Our paper primarily focuses on the performance-structure correlation near the contact interface between TIMs and electronic components. To clarify this distinction, in revised manuscript we use the term 'thermal contact resistance' specifically refers to the interface between TIM and electronics, while the term 'total thermal resistance' discussing the TIM as the whole interface between electronics and heatsink.

Following the reviewer's advice, we have enhanced the morphological characterization of the actual interface between the composite TIMs and electronic components, as presented in Figure R3. For the substrate, it refers to the metal plate from the standard equipment (LW9389, Longwin, Taiwan). Regarding the TIMs, we observed that horizontally aligned BNNS are present on the top layer across all BNNS-TIM samples with varying filler loadings, as shown in Figure 1C. Furthermore, we conducted surface roughness measurements of the BNNS-TIMs at different filler loadings, employing 3D analysis with a profilometer (Tencor P7, KLA co., US). The samples exhibited similar surface roughness, with an average surface roughness around 5 μm , indicating minimal influence of surface roughness on the thermal resistance results. These results has been added in the main text (Page 10) as follows: "Considering that the composite films with 70-90 wt.% BNNSs exhibit a consistent surface morphology with a thin layer of horizontally aligned BNNS (Figure 1C), along with close surface hardness of 92-96 (Shore A) (Figure 4A) and similar surface roughness of around 5 μm (Figure S7C), it is surmised that the interfacial microstructure of the TIMs plays a key role in the interfacial phonon transport."

Figure R3: Surface roughness (Sa, Arithmetic Mean Deviation) of BNNS-TIMs across various filler loadings ranging from 50 to 90 wt.%. Inset at the top-right corner is a 3D reconstructed image of a 70 wt.% BNNS-TIM sample surface.

Finally, I am confused by the lack of information about the BNNS: their dimensions, how smooth or wrinkled they are, as well as about the polymer: it is apparently a thermoplastic but what is its chemistry? is it binding to the BNNS?

Response: Thank you for your valuable inquiry regarding our work. In accordance with the reviewer's guidance, we have provided the detailed information as follows:

1. BNNS Dimensions: The distribution on lateral size of the BNNS utilized in this study is obtained from a statistical analysis of 130 nanosheets by SEM (Figure R4A). The average lateral size is around 1.1 μm. And for thickness is obtained from a statistical analysis of 50 nanosheets by AFM (Figure R4B). The average thickness of BNNSs is 1.56 nm, indicating the 4-5 layer thickness. For a comprehensive understanding of the BNNS dimensions, please refer to our previous work². The production details of BNNSs has been added in the main text (Page 14).

Figure R4. (A-B) Statistical analysis of BNNS dimensions. (A) Lateral size of BNNS, and (B) Thickness distribution of BNNS. (C) FTIR spectra of BNNS-TIM at 70 wt.% loading, polymer matrix, and BNNS.

2. BNNS Morphology: The Raman spectroscopy data presented in Figure R5 indicate negligible Raman shift of BN in both our BNNS-TIMs and the pure BNNS powder. This suggests that the majority of BNNS in our study are straight and unstrained, and the existence of wrinkled BNNSs could be minimal. The Raman characterization result has been updated into Figure S1F and the method has been described in the Characterization section in the main text (Page 15).

Figure R5: Raman Spectra of BNNS-TIMs and Raw h-BN Powder. This figure illustrates the Raman spectra captured when the laser beam targets the cutting face of BNNS-thermal interface materials (BNNS-TIMs), where the BNNS are aligned parallel (C1// & C2//), and the internal region, where the BNNS are oriented perpendicular (C1⊥ & C2⊥). The Raman spectrum of raw h-BN powder is provided as a reference.

3. Polymer Chemistry: The polymer matrix employed is an acrylates polymer characterized by a low molecular weight ($M_n \sim 10^3$ - 10^4), as stated in Page 14 of the main text.

4. Polymer-BNNS Interaction: Fourier Transform Infrared Spectroscopy (FTIR) analysis reveals a minor red-shift of the characteristic BNNS peaks within the composite (Figure R4C). Specifically, the in-plane stretching mode and the out-of-plane bending mode both of B-N bonding are observed at 1369 cm^{-1} and 811 cm^{-1} , respectively for BNNS³. A red-shift of approximately 15 cm^{-1} is noted in the composite (BNNS-TIM), indicating non-covalent interaction⁴ between BNNS and the polymer. And no observation of new peaks suggesting the absence of newly formed covalent bonds. The FTIR result has been incorporated into Supporting Information as Figure S1E (Page 3), with corresponding description in the main text (Page 4) as follows: “The infrared spectroscopy analysis (Figure S1) suggested the non-covalent interaction between BNNS and the polymer.”

Reviewer #3

In this work, the authors demonstrated that the interfacial phonon bridge is an effective strategy to optimize the thermal resistance within the hexagonal boron nitride nanosheets based composite

films. With this method, the authors reported that the contact resistance can be significantly reduced by 70%. The results of this work show promising applications in the field of thermal interface materials (TIMs). However, due to the following issues listed below, I suggest the authors should adequately address the following comments to significantly improve this manuscript before potential publication:

Response: We deeply appreciate the reviewer's exceptionally professional and highly constructive feedback on our manuscript.

Comment 1. The recent advances on the thermal properties of two-dimensional materials [Applied Physics Reviews, 10, 041404 (2023); Physical Review B, 107, 165424 (2023)] and interfacial thermal transport [Nanotechnology, 33, 035707 (2022)] are highly related to this work, and thus should be cited in the introduction part to provide a timely survey of literature studies for the readers.

Response: We thank the reviewer's recommendation, and these papers are extremely valuable, offering profound insights into the thermal transport behaviors in graphene materials. We have incorporated them as references 20-23 into the introduction part of our paper (Page 3).

Comment 2. The interfacial thermal resistance between Van der Waals stacking 2D materials in the out-of-plane direction is a prevalent issue in thermal management. In fact, the concept of phonon bridging is not new for interfacial optimization, see for instance Nat. Commun. 7, 11281 (2016). I suggest the authors should moderate its tone. The advantage for optimization simply by cutting operation would be a more intriguing focus for this manuscript.

Response: We sincerely appreciate the reviewer's reminder regarding the originality of the phonon bridge concept. Acknowledging the prior use of this concept in the field, we have revised our manuscript to reflect a more moderate tone and have incorporated relevant citations (see it in comment 1). We recognize that previous studies have primarily focused on fundamental atomic-level investigations. We have highlighted the literatures mentioned in comment 1 and comment 2 as follows: "Recent studies by Chen, Volz and co-workers have shown that the interfacial phonon transport among 2D nanosheets could be mediated by their stacking structures and binding modes." Indeed, our study's key innovation lies in translating these nanoscale thermal advantages into bulk systems, leading to a low thermal resistance for practical applications.

Comment 3. The arc-like structure resulting from cutting is observed only at the surface of BNNS-TIMs. How would the optimization performance of thermal contact resistance R vary with the size of the sample? With larger BNNS-TIM structures, the impact of surface in principle should eventually become negligible. Would this effect potentially diminish the optimization efficiency of R from surface cutting ?

Response: This aspect was indeed a consideration in our research. Our results show that as the thickness of the 70 wt.% BNNS-TIMs sample increases from 200 μm to 1 mm, the proportion of the surface layer to the overall structure decreases from $\sim 25\%$ to $\sim 5\%$ (Figure S7). Meanwhile, the ratio of thermal contact resistance to total thermal resistance decreases from $\sim 80\%$ to $\sim 50\%$ (Figure R6). This indicates reduced significance of surface cutting in optimizing thermal resistance as sample thickness increases. In practical applications of TIMs, a smaller thickness (less than 1 mm) is typically preferred, provided that adequate dielectric performance is maintained⁵. The ratio of thermal contact resistance to total thermal resistance would be larger than $\sim 50\%$, and the optimization of thermal resistance via surface modification remains highly effective.

Figure R6: Total thermal resistance of 70 wt.% BNNS-TIMs at thickness ranging from 0.2 to 1.6 mm, under pressure of 50 psi. The linear fitting indicates the thermal contact resistance and effective thermal conductivity.

Comment 4. The authors attempted to use the contact angle to elucidate the myography dependent on filler content and the resulting changes in R . The second paragraph on page 10 implied that samples with high filler content, such as 90 wt.%, exhibit high contact angles associated with lower thermal conductance and higher R . However, Figures 1(f) and 2(c) indicated no angle difference for BNNS at the surface. The provided theoretical explanation may lack

accuracy.

Response: After carefully reviewing the reviewer's comment, we think the reviewer want to refer to Figure 1c (SEM image of BNNS-TIMs) rather than Figure 2c (FEM simulation results).

In our study, the surface layer of BNNSs across all BNNS-TIMs with varying filler loadings exhibited a consistent horizontal arrangement. This horizontal layer acts as an effective heat spreader, enhancing the contact area between the TIM and the heatsink or chip. Such arrangement transforms the line contact into a more efficient plane contact, as opposed to direct contact of perpendicular BNNSs with the heatsink or chip⁶. Aligned with these experimental findings, our MD model consistently defines the surface horizontal layer of BNNSs on both sides, and we focus on assessing and optimizing phonon diffusion between internal perpendicular BNNSs and the surface horizontal BNNSs. We are focusing on the interfacial thermal transport between BNNSs at contact of different angle, while for the BNNS at TIM surface contacting the metal (electronics or heatsink) is not included in our theoretical explanation. The focus on BNNS interconnection has been prominently stressed in the main text (Page 10).

Comment 5. The authors employed the overlap of phonon density of states (DOS) to explain the interfacial thermal transport. This approach might be oversimplified for comprehending the complicated interfacial thermal transport mechanism, at least for the papers in the Nat. Commun. level. The recent advancements in theoretical and computational methods offer numerous alternative approaches to elucidate interfacial thermal transport and provide more detailed insights into phonon transport mechanisms, such as the interfacial phonon transmission spectrum analysis [see for instance, Nano Letters, 21, 2634–2641 (2021)]. I suggest the authors should strength the theoretical analysis part.

Response: Thanks very much for the reviewers' recommendation. Accordingly, we have integrated the approach of interfacial phonon transmission spectrum analysis, a method gaining prominence in phonon transport studies^{7,8}, into our research framework. Our calculations focus on the phonon transmission spectrum specifically at the contact areas of BNNSs. We observed that the transmission function of interfacial phonons is notably higher when BNNS are positioned at a contact angle of 30° compared to larger angles, diminishing progressively as the contact angle increases (Figure R7). This effect is more pronounced for low-frequency phonons (<10 THz), which play a major role in interfacial thermal transport^{9,10}. Correspondingly, there is a notable decrease in thermal conductance beyond the 30° angle, as illustrated in Figure 4E. This finding is pivotal in our study as it establishes that smaller contact angles are instrumental in enhancing

phonon transport efficiency, subsequently reducing thermal resistance at BNNS interfaces. Figure R7 has been incorporated as Figure 4F in the main text, with corresponding discussion (Page 10-12), and the calculation details (Page 16-17).

Figure R7: Interfacial phonon transmission spectrum for varying BNNS contact angles. This figure presents the transmission spectrum of interfacial phonons in BNNS at contact angles of 30°, 45°, 60°, and 90°. The y-axis represents the spectral transmission function, $\Gamma(\omega)$, and the x-axis corresponds to the phonon frequency.

Comment 6. Figure 1(f) also indicated that the BNNS likely exhibit a curved surface after cutting, which may potentially cause a partial shift from out-of-plane thermal transport to in-plane transport. However, the model used in the theoretical explanation did not account for this notable feature.

Response: We sincerely appreciate the reviewer’s insightful observation regarding the potential influence of curved surfaces on the thermal transport properties of BNNS in our study. Firstly, we concur with the reviewer’s assessment that the curved surface of BNNS may result in a hybrid direction for phonon transport, we also acknowledge, in line with existing literature^{11–13}, that the strain induced by such curvature can significantly impact the thermal behavior of two-dimensional materials.

To address this concern, we characterize the strain of the BNNS in both the cutting face and the internal part of the BNNS-TIMs using Raman spectroscopy. We specifically focused on the E_{2g} phonon mode peak ($\sim 1366\text{ cm}^{-1}$), renowned for its strain sensitivity, distinct intensity, and sharpness in h-BN Raman spectrum^{14,15}. As delineated in Figure R5 (see it in reviewer 2’s comment), our analysis revealed negligible peak shifts in the BNNS, both on the cutting face and

within the internal part of the BNNS-TIMs, when compared to raw h-BN powder.

Based on these findings, we infer that the BNNS incorporated in our BNNS-TIMs are negligibly strained, implying an absence of significant curvature in the BNNS themselves. We attribute the curvature observed in Figure 1(f) primarily to unsupported BNNS-composite layers at the fracture face of the sample used for morphological characterization via SEM. In the context of the entire BNNS-TIM structure, we observe that curvature predominantly occurs at BNNS interconnections with different contact angles, while individual BNNS remain flat. Consequently, we believe our model is reasonable by setting the initial strain of BNNS to zero in our modeling, which is consistent with our experimental observations, that the strain effects from curvature are minimal in our BNNS-TIMs. The negligible strained state of BNNS in composite is stated in the main text (Page 10). The Raman characterization method has been added in the Characterization section in the main text (Page 15).

References

1. Min, S., Blumm, J. & Lindemann, A. A new laser flash system for measurement of the thermophysical properties. *Thermochim Acta* **455**, 46–49 (2007).
2. Zhou, Y. *et al.* Viscous Solvent-Assisted Planetary Ball Milling for the Scalable Production of Large Ultrathin Two-Dimensional Materials. *Acs Nano* **16**, 10179–10187 (2022).
3. Geick, R., Perry, C. H. & Rupprecht, G. Normal Modes in Hexagonal Boron Nitride. *Phys. Rev.* **146**, 543–547 (1966).
4. Wang, Y. ying *et al.* Raman Studies of Monolayer Graphene: The Substrate Effect. *J. Phys. Chem. C* **112**, 10637–10640 (2008).
5. Chung, D. D. L. Performance of Thermal Interface Materials. *Small* **18**, 2200693 (2022).
6. Wu, K., Liu, D., Lei, C., Xue, S. & Fu, Q. Is filler orientation always good for thermal management performance: A visualized study from experimental results to simulative analysis. *Chem Eng J* **394**, 124929 (2020).
7. Ren, W. *et al.* The Impact of Interlayer Rotation on Thermal Transport Across Graphene/Hexagonal Boron Nitride van der Waals Heterostructure. *Nano Lett.* **21**, 2634–2641 (2021).
8. Islam, A. S. M. J., Islam, Md. S., Ferdous, N., Park, J. & Hashimoto, A. Vacancy-induced thermal transport in two-dimensional silicon carbide: a reverse non-equilibrium molecular dynamics study. *Phys. Chem. Chem. Phys.* **22**, 13592–13602 (2020).
9. Sachat, A. E. *et al.* Effect of crystallinity and thickness on thermal transport in layered PtSe₂. *npj 2D Mater. Appl.* **6**, 32 (2022).
10. Qiu, L., Zhang, X., Guo, Z. & Li, Q. Interfacial heat transport in nano-carbon assemblies. *Carbon* **178**, 391–412 (2021).
11. Nakagawa, K. *et al.* Controlling the thermal conductivity of multilayer graphene by strain. *Sci. Rep.* **11**, 19533 (2021).
12. Dheeraj, K. V. S. & Sathian, S. P. The disparate effect of strain on thermal conductivity of 2-D materials. *Phys. Chem. Chem. Phys.* **23**, 23096–23105 (2021).
13. Ding, B., Li, X., Zhou, W., Zhang, G. & Gao, H. Anomalous strain effect on the thermal conductivity of low-buckled two-dimensional silicene. *Natl. Sci. Rev.* **8**, nwaa220 (2020).
14. Mohiuddin, T. M. G. *et al.* Uniaxial strain in graphene by Raman spectroscopy: G peak splitting, Grüneisen parameters, and sample orientation. *Phys. Rev. B* **79**, 205433 (2009).
15. Wang, W., Li, Z., Marsden, A. J., Bissett, M. A. & Young, R. J. Interlayer and interfacial stress transfer in hBN nanosheets. *2D Mater.* **8**, 035058 (2021).

REVIEWER COMMENTS

Reviewer #1 (Remarks to the Author):

The authors have answered most of my questions. TIMs with ultralow interfacial thermal resistance are prepared and their manufacturability is demonstrated in this manuscript. However, the blade coating, multi-layer compression, and cutting processes for preparing TIMs are quite routine. The performance achieved in this manuscript is mainly attributed to the design of the acrylate polymer. However, information about the polymer is quite limited. The results presented in this manuscript do not provide enough novelty to merit publication in Nature Communications. Given that the authors will continue this work, the following comments are provided to improve its content:

1. The authors should provide more information about the polymer. For example, the structural formula of the polymer should be added in the manuscript.
2. The authors have demonstrated the tensile properties of their composites with different BN loadings. In fact, the compression properties are more important as TIMs generally undergo packaging pressure in practical application. The authors should provide the compression properties of the pure polymer and their composites with different BN loadings.
3. There are few functional groups on the BNNS surface, which may lead to poor compatibility with polymer. Did the authors functionalize the BNNS to ensure the compatibility between the BNNS and the polymer?

Reviewer #2 (Remarks to the Author):

The authors addressed most of my comments in a satisfying way except regarding the polymer. It is desirable to have more precise information about this acrylate: where was it purchased, what is the chemical formula, how was it processed, etc.

Reviewer #3 (Remarks to the Author):

The authors have adequately addressed all the comments in the revision. This work is now suitable for publication.

Editorial Note: Parts of this Peer Review File have been redacted as indicated to maintain the confidentiality of trade secrets.

Response to Reviewer's Comments

Dear Editors and Reviewers,

We would like to thank you all for your valuable feedback on our revised manuscript. We have reviewed the comments carefully and revised our manuscript accordingly. Our responses to each comment have been detailed below, and the revisions are clearly marked in blue in the updated version of the manuscript.

Reviewer #1

The authors have answered most of my questions. TIMs with ultralow interfacial thermal resistance are prepared and their manufacturability is demonstrated in this manuscript. However, the blade coating, multi-layer compression, and cutting processes for preparing TIMs are quite routine. The performance achieved in this manuscript is mainly attributed to the design of the acrylate polymer. However, information about the polymer is quite limited. The results presented in this manuscript do not provide enough novelty to merit publication in Nature Communications. Given that the authors will continue this work, the following comments are provided to improve its content:

Response: We are grateful for the reviewer's recognition that most of the previous inquiries have been addressed. We apologize for the oversight in not providing detailed information about the polymers in our revised manuscript. This occurred because our fabricated TIMs with ultralow interfacial thermal resistance have garnered significant interest from industrial companies (*i.e.*, vivo, Sunwoda, HuaWei and BYD, some are the key authors of this work), and are progressing towards the commercialization.

The reviewer has noticed the pivotal role of the acrylate polymer in enhancing the performance of TIMs, which we fully agree. As detailed in response to your previous comments, specifically comment 2 and 3, we have illustrated how the low molecular weight of this polymer imparts substantial viscoplasticity to the BNNS/polymer composites, inducing the rotation of BNNS near the cutting surfaces to form an arc-like structure and thereby reducing the interfacial contact resistance. In light of both Reviewer 1 and Reviewer 2's requests for detailed polymer information, we have decided to disclose the comprehensive details of the polymer to elucidate the reproducibility, as outlined in the main text (Page 14): "[Redacted] and the structural formula is

provided in Figure S1G. The dispersion of BNNS was first blended with the acrylate polymer at a low molecular weight ($M = 384.5$)."

We also understand the reviewer's concern regarding the potential impact on the novelty of our study due to the previously employed fabrication techniques. However, it's crucial to emphasize that our work primarily focuses on the exploration and transformation of nano-scale thermal phenomena in 2D materials into bulk systems to achieve low thermal resistance for practical TIMs applications, rather than innovating the fabrication process itself. The novelty and insights generated from our work have been clarified in our previous response and were agreed upon by Reviewers 2 and 3. As this relevant information was not included in the previous reply to the reviewer, we are including it here for your reference.

"We agree with the reviewer's comments regarding prior strategies utilizing shear to align nanosheets and the significance of vertical alignment of BNNS in enhancing thermal conductivity, as discussed in our introduction. However, the overall thermal resistance of thermal interface materials (TIMs) encompasses not only the inherent thermal resistance of the materials but also the thermal contact resistance. Although vertical alignment of BNNS can enhance the thermal resistance of the materials themselves, it may introduce substantial thermal contact resistance between TIMs and the substrate due to directional phonon transport along 2D nanosheets. In this work, we focus on optimizing interfacial thermal transport by tilted BNNSs contact, distinct from strategies employing solely the vertically aligned nanosheets. While we employ previously reported fabrication approach mentioned by the reviewer (*i.e.*, layer stacking and shear cutting) to tune the orientation of BNNS, our key discovery is the contact-dependent thermal resistance and firstly demonstrate how this nanoscale phonon bridge effect can be translated into bulk systems to enable high-performance TIMs. Overall, this study provides expands the knowledge of the thermal properties of bulk 2D-material assemblies and offers a realistic approach to the industrialization of 2D materials-based dielectric TIMs."

More specifically, the novel aspects of our research are summarized as follows:

1. Through the application of an acrylate polymer matrix and proper cutting techniques, we can manipulate the BNNS orientation to create a unique arc-like structure that significantly reduces the thermal resistance. This discovery is a novel thermal resistance phenomenon, diverging from previous TIMs studies that primarily concentrate on the vertical alignment of 2D materials to amplify through-plane thermal conductivity, often overlooking the importance of the interface contact resistance.
2. The non-equilibrium molecular dynamics simulations revealed that oblique BNNSs bridged the

phonon spectra mismatch between nanosheets aligned parallel and perpendicular to the films, leading to a significant reduction in the possibility of back-reflection of phonons at the boundary and an improved efficiency of phonon coupling. This provides theoretical insights that deepen our understanding of the intricate interplay between phonon transport and 2D nanosheet arrangement.

3. This unique arc-like structure of BNNS-TIM has enabled unexpected low thermal resistance. As compared in Table S4, the thermal and electric performance of BNNS-TIMs significantly outperforms that of the representatives of commercially available dielectric TIMs.. Indeed, the utilization of a 'routine' fabrication strategy proves advantageous for the large-scale manufacturing of TIMs, given its high compatibility with existing industrial processes. This facilitates the practical application of TIMs using the nanoscale phonon bridge effect in bulk systems. For instance, we have closely collaborated with Sunwoda, global leaders in the lithium-ion battery industry, to explore potential applications of our TIMs in fast-charging batteries. Additionally, we are establishing close partnerships with Huawei and BYD to assess the suitability of our composites for cooling Insulated Gate Bipolar Transistors (IGBTs) in electric cars.

Overall, this work contributes substantially to the understanding of thermal properties in bulk 2D material assemblies and opens new possibilities for their industrial application in advanced dielectric TIMs. We believe that the novelty and impact of our research support its potential publication in Nature Communications.

Comment 1. The authors should provide more information about the polymer. For example, the structural formula of the polymer should be added in the manuscript.

Response: We apologize for any confusion caused by the initial omission. [Redacted] The structural formula is shown in Figure R1 and has been incorporated in the supporting information as Figure S1G (Page 3) and relevant texts are added in the main text (Page 6&14).

Figure R1: Structural formula of polymer matrix which is an acrylic acid-butyl acrylate-2-ethylhexyl acrylate copolymer.

Comment 2. The authors have demonstrated the tensile properties of their composites with different BN loadings. In fact, the compression properties are more important as TIMs generally undergo packaging pressure in practical application. The authors should provide the compression properties of the pure polymer and their composites with different BN loadings.

Response: Thank you for highlighting the importance of compression properties in practical applications of TIMs. We have included the compression properties of the composites with various BNNS loadings in Figure S2B of the supporting information (Page 4), and have now incorporated the compression properties of the pure polymer into Figure S2B, depicted as Figure R2. According to the compressive data, the fabricated TIMs could effectively maintain the shape fidelity when subjected to packaging pressure (~50 or 100 psi) in real-world applications, as clarified in the main text (Page 9).

Figure R2: Stress-strain curves describing the compressive behavior of BNNS-TIMs at loadings of 50-90 wt.% and pure polymer matrix.

Comment 3. There are few functional groups on the BNNS surface, which may lead to poor compatibility with polymer. Did the authors functionalize the BNNS to ensure the compatibility between the BNNS and the polymer?

Response: Thanks for the reviewer’s insightful question. Indeed, the BNNS surface is functionalized with limited hydroxyl groups (e.g. -OH), as demonstrated in our previous works^{1,2}. To address the reviewer's query regarding the compatibility between BNNS and the polymer, we have extensively reviewed literature on BNNS/polymers. Our understanding is as follows:

The polarized nature of the B-N bonds and the intrinsic high bonding potential of B and N atoms that significantly bolster the interaction between BN and various molecules, despite the BN surface being nearly atomically smooth³⁻⁶. The polarization yields augmented electrostatic energy, and the strong bonding potential yields high van der Waals energy. As a result, the BN-molecule interaction is markedly greater than that of non-polarized carbon-molecule interaction, which has been verified by experimental data and MD simulations for various molecules, ranging from large (e.g., PMMA, epoxy)^{3,4} to small ones (e.g., water)^{6,7}. In some cases, BNNS also demonstrates interfacial adhesion with polymers that is comparable to that of graphene oxide (GO), which is known for its functional groups⁸. Thus, despite its limited functionalization, BNNS could establish a compatible and robust interface with polymers.

To clarify this, we added the sentence "The BNNSs were mixed with an acrylate polymer to create a compatible dispersion through the strong molecular interactions between the polarized B-N bond and the polymer." in the main text (Page 4).

Reference

1. Zhou, Y. *et al.* Viscous Solvent-Assisted Planetary Ball Milling for the Scalable Production of Large Ultrathin Two-Dimensional Materials. *Acs Nano* **16**, 10179–10187 (2022).
2. Xu, L. *et al.* A Malleable Composite Dough with Well-Dispersed and High-Content Boron Nitride Nanosheets. *Acs Nano* **17**, 4886–4895 (2023).
3. Chen, X. *et al.* Mechanical strength of boron nitride nanotube-polymer interfaces. *Appl. Phys. Lett.* **107**, 253105 (2015).
4. Rouhi, S. Molecular dynamics simulation of the adsorption of polymer chains on CNTs, BNNTs and GaNNTs. *Fibers Polym.* **17**, 333–342 (2016).
5. Rasul, M. G., Kiziltas, A., Arfaei, B. & Shahbazian-Yassar, R. 2D boron nitride nanosheets for polymer composite materials. *Npj 2d Mater Appl* **5**, 56 (2021).
6. Tocci, G., Joly, L. & Michaelides, A. Friction of Water on Graphene and Hexagonal Boron Nitride from Ab Initio Methods: Very Different Slippage Despite Very Similar Interface Structures. *Nano Lett.* **14**, 6872–6877 (2014).
7. Secchi, E. *et al.* Massive radius-dependent flow slippage in carbon nanotubes. *Nature* **537**, 210–213 (2016).
8. Pinto, G. M. *et al.* Exploring the relationship between interfacial adhesion, molecular dynamics, and the brill transition in fully bio-based polyamide 1010 nanocomposites reinforced by two-dimensional materials. *Polymer* **289**, 126482 (2023).

Reviewer #2

The authors addressed most of my comments in a satisfying way except regarding the polymer. It is desirable to have more precise information about this acrylate: where was it purchased, what is the chemical formula, how was it processed, etc.

Response: Thank you for your acknowledgement of our effort in addressing your comments. The polymer, identified by the CAS number 26710-97-4, is commercially available from LEAP CHEM CO., LTD., with a low molecular weight (384.5 g mol⁻¹). For comprehensive details on its chemical composition, the chemical formula (C₁₁H₂₀O₂·C₇H₁₂O₂·C₃H₄O₂)_x can be referenced via its CAS number. For the processing, the polymer is directly blended with BNNS dispersion as stated in the main text (Page 14). We have corrected and updated this information in the main text (Page 6&14) and the supporting information (Page 3) as follows

"The polymer matrix was an acrylate polymer with the CAS number as 26710-97-4 and the chemical formula (C₁₁H₂₀O₂·C₇H₁₂O₂·C₃H₄O₂)_x. It was supplied by LEAP CHEM CO., LTD. (product number: B0508771), and the structural formula is provided in Figure S1G. The dispersion of BNNS was first blended with the acrylate polymer at a low molecular weight (M = 384.5)."

Reviewer #3

The authors have adequately addressed all the comments in the revision. This work is now suitable for publication.

Response: We are immensely grateful for your thorough review and positive feedback on our manuscript's revision. We sincerely thank you for your constructive comments and guidance throughout the revision process.

REVIEWERS' COMMENTS

Reviewer #2 (Remarks to the Author):

I am satisfied with the last revision made by the authors to address my comment.

Response to Reviewer's Comments

Dear Editors and Reviewers,

We would like to thank you all for your valuable feedback on our revised manuscript. Our response to the comment has been detailed below.

Reviewer #2 (Remarks to the Author):

I am satisfied with the last revision made by the authors to address my comment.

Response: We sincerely appreciate your positive feedback regarding our research.